# CROSS-DOMAIN OFFLINE POLICY ADAPTATION WITH OPTIMAL TRANSPORT AND DATASET CONSTRAINT

**Jiafei Lyu**[1] [*]**Mengbei Yan**[1]**, Zhongjian Qiao**[1]**, Runze Liu**[1]**, Xiaoteng Ma**[2]**, Deheng Ye**[3]**,**
**Jingwen Yang**[3]**, Zongqing Lu**[4,5]**, Xiu Li**[1,†]

[1]Tsinghua Shenzhen International Graduate School, Tsinghua University
[2]Department of Automation, Tsinghua University, [3]Tencent
[4]School of Computer Science, Peking University, [5]Beijing Academy of Artificial Intelligence
`lvjf20@mails.tsinghua.edu.cn, li.xiu@sz.tsinghua.edu.cn`

## ABSTRACT

We explore cross-domain offline reinforcement learning (RL) where offline datasets from another domain can be accessed to facilitate policy learning. However, the underlying environments of the two datasets may have dynamics mismatches, incurring inferior performance when simply merging the data of two domains. Existing methods mitigate this issue by training domain classifiers, using contrastive learning methods, etc. Nevertheless, they still rely on a large amount of target domain data to function well. Instead, we address this problem by establishing a concrete performance bound of a policy given datasets from two domains. Motivated by the theoretical insights, we propose to align transitions in the two datasets using optimal transport and selectively share source domain samples, without training any neural networks. This enables reliable data filtering even given a few target domain data. Additionally, we introduce a dataset regularization term that ensures the learned policy remains within the scope of the target domain dataset, preventing it from being biased towards the source domain data. Consequently, we propose the Optimal Transport Data Filtering (dubbed OTDF) method and examine its effectiveness by conducting extensive experiments across various dynamics shift conditions (*e.g.*, gravity shift), given limited target domain data. It turns out that OTDF exhibits superior performance on many tasks and dataset qualities, often surpassing prior strong baselines by a large margin.

## 1 INTRODUCTION

Alice used to play tennis without any exposure to other ball sports. Recently, the tennis court needs maintenance, and Alice ends up playing badminton with Bob. Alice quickly gets familiar with this sport. As depicted in this example, human beings are capable of swiftly adapting their policies to structurally similar tasks. We expect the same phenomenon to be observed in reinforcement learning (RL) agents. To that end, we aim at achieving better performance in the *target domain* with a limited budget by accessing a *source domain* (*e.g.*, a simulator) where the two domains may have distinct transition dynamics. Such a setting is referred to as the *policy adaptation* problem (Xu et al., 2023).

Many works focus on handling the online policy adaptation problem (Xu et al., 2023; Lyu et al., 2024a; Niu et al., 2022) where either the source domain or the target domain is online. Instead, we are interested in *offline policy adaptation*, or the cross-domain offline RL problem (Wen et al., 2024) (*i.e.*, both domains are offline), since online interactions can be expensive and even dangerous. The cross-domain offline RL setting is common in practice. For example, one research team gathered historical trajectories of a humanoid robot. However, over time, the robot's physical body parts may degrade or get damaged during operation. That said, the dataset collected by then would differ from the past datasets in transition dynamics. It becomes a typical cross-domain offline RL scenario if one decides to utilize past data for training policies. Recent advances in cross-domain offline RL include learning domain classifiers to estimate the dynamics gap (Liu et al., 2022a), filtering source domain data based on mutual information (Wen et al., 2024), etc. Unfortunately, these methods still run on

---

[*]Work done while working as an intern at Tencent. [†] Corresponding Author.

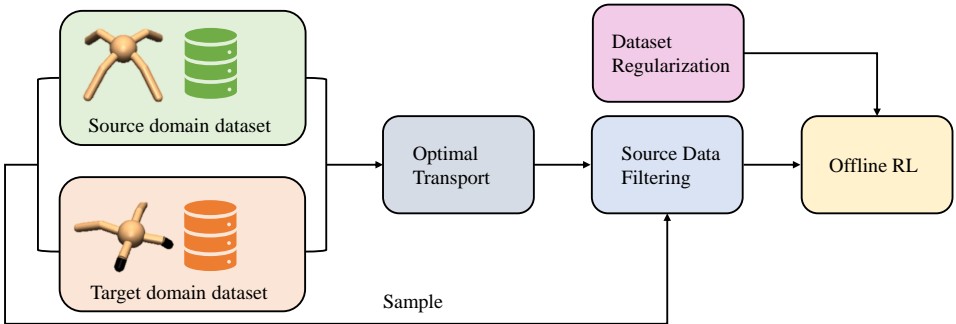

Figure 1: **An overview of our proposed framework.** We first align source domain data and target domain data via the Wasserstein distance. Then we adopt the solved optimal coupling for selectively sharing source domain transitions with the downstream offline RL algorithms. We further introduce a regularization term to encourage the learned policy to lie in the support region of the target domain.

a comparatively large target domain dataset. This can be problematic since offline RL methods like ReBRAC (Tarasov et al., 2024) can achieve strong performance on it, downgrading the necessity of an extra source domain dataset. Hence, we consider the offline policy adaptation problem given very limited target domain data, with which the single-domain offline RL methods often struggle.

In light of such a challenge, we first theoretically characterize the performance bound of a policy in the true target domain and the empirical source domain, which is related to the deviations of the learned policy against the behavior policies in the source domain dataset and the target domain dataset, and the dynamics gap between the two domains. The theoretical results highlight the necessity of selectively sharing source domain data to close the performance gap rather than simply merging data from two domains. Given that only a few target domain data are accessible, we resort to *optimal transport* to find optimal alignments between data from two domains without training any neural network. We then use the similarity measure between a transition in the source domain dataset and the entire target domain dataset to filter out dissimilar source domain data. This enables reliable source domain data selection regardless of the target domain dataset size. However, it is still insufficient to fulfill efficient offline policy adaptation, since the performance bound is also controlled by the deviation of the learned policy against data-collecting policies in both domains. To mitigate this, we further introduce a policy regularization term that constrains the learned policy from executing out-of-distribution (OOD) actions outside the support region of the target domain dataset. We name our method Optimal Transport Data Filtering (OTDF) and summarize its framework in Figure 1.

We evaluate OTDF upon various D4RL (Fu et al., 2020) datasets with different types of dynamics shifts (*e.g.*, gravity shift), given limited target domain data. Empirically, we demonstrate that OTDF achieves superior performance across numerous tasks and with varied source or target domain dataset qualities, often outperforming recent strong baseline methods by a large margin. To ensure that our work is reproducible, our code is available at https://github.com/dmksjfl/OTDF.

## 2 PRELIMINARIES

**Reinforcement Learning (RL).** RL problems can be formulated by a Markov Decision Process (MDP) $\mathcal{M}$, which is defined by the 5-tuple $\mathcal{M} = (\mathcal{S}, \mathcal{A}, P, r, \gamma)$ where $\mathcal{S}$ denotes the state space, $\mathcal{A}$ is the action space, $P$ represents the transition probability, $r(s, a) : \mathcal{S} \times \mathcal{A} \to \mathbb{R}$ is the scalar reward signal, and $\gamma \in [0, 1)$ is the discount factor (Sutton & Barto, 2018). We assume that the reward signals are bounded, *i.e.*, $|r(s, a)| \leq r_{\max}, \forall s, a$. The objective of the RL agent is to learn a policy $\pi$ to maximize the expected discounted cumulative return $\mathbb{E}_\pi[\sum_{t=0}^\infty \gamma^t r(s_t, a_t)]$. We assume that we have access to a *source domain* $\mathcal{M}_{\mathrm{src}} = (\mathcal{S}, \mathcal{A}, P_{\mathrm{src}}, r, \gamma)$ and a *target domain* $\mathcal{M}_{\mathrm{tar}} = (\mathcal{S}, \mathcal{A}, P_{\mathrm{tar}}, r, \gamma)$, where the two domains only differ in their transition dynamics. We denote the normalized probability that a policy $\pi$ encounters the state action pair $(s, a)$ in the domain $\mathcal{M}$ as $\rho_\mathcal{M}^\pi(s, a) := (1 - \gamma) \sum_{t=0}^\infty \gamma^t P_{\mathcal{M},t}^\pi(s) \pi(a|s)$ where $P_{\mathcal{M},t}^\pi(s)$ denotes the probability that the policy $\pi$ encounters the state $s$ at timestep $t$ in $\mathcal{M}$. Then, the performance of a policy $\pi$ in $\mathcal{M}$ can be formulated as $J_\mathcal{M}(\pi) = \mathbb{E}_{s,a\sim\rho_\mathcal{M}^\pi}[r(s, a)]$. We denote $P_{\mathrm{src}} = P_{\mathcal{M}_{\mathrm{src}}}, P_{\mathrm{tar}} = P_{\mathcal{M}_{tar}}$.

In offline RL, the agent can only get access to a static dataset $D = \{(s_i, a_i, r_i, s_{i+1})\}_{i=1}^N$, where $N = |D|$ is the dataset size. The goal of cross-domain offline RL is to improve the performance of the agent in the target domain by leveraging the mixed dataset $D_{\text{mix}} = D_{\text{src}} \cup D_{\text{tar}}$, where $D_{\text{src}}$ is the source domain dataset and $D_{\text{tar}}$ is the target domain dataset. We further define the empirical MDP in the dataset $D$ as $\widehat{\mathcal{M}} := (\mathcal{S}, \mathcal{A}, \widehat{P}, r, \gamma)$. $\widehat{P}$ denotes the empirical transition probability in the dataset and $\widehat{P}(s'|s, a) = 0$ for all OOD transition pairs. That said, we have two empirical MDPs in cross-domain offline RL, $\widehat{\mathcal{M}}_{\text{src}}$ with dynamics $P_{\widehat{\mathcal{M}}_{\text{src}}}$, $\widehat{\mathcal{M}}_{\text{tar}}$ with dynamics $P_{\widehat{\mathcal{M}}_{\text{tar}}}$.

**Optimal Transport (OT).** OT (Cuturi, 2013; Peyré & Cuturi, 2019) is a popular method to measure distribution discrepancy. The Wasserstein distance between two discrete measures $\mu_x = \frac{1}{n}\sum_{t=1}^n \delta_{x_t}$ and $\mu_y = \frac{1}{n'}\sum_{t=1}^{n'} \delta_{y_t}$ is defined as:

$$\mathcal{W}(\mu_x, \mu_y) = \min_{\mu \in M} \sum_{t=1}^n \sum_{t'=1}^{n'} C(x_t, y_{t'})\mu_{t,t'}, \tag{1}$$

where $C$ is a cost function, $M = \{\mu \in \mathbb{R}^{n \times n'} : \mu\mathbf{1} = \frac{1}{n}\mathbf{1}, \mu^T\mathbf{1} = \frac{1}{n'}\mathbf{1}\}$ is the set of the coupling matrices, $\delta_x$ denotes a Dirac delta measure for $x$, $\mu_{t,t'}$ is the $t$-th row, $t'$-th column element in $\mu$, and $n, n'$ are the sizes of the measures $\mu_x, \mu_y$, respectively. Solving Equation 1 results in an optimal coupling $\mu^*$ that depicts an alignment between the samples in $\mu_x$ and $\mu_y$.

## 3 CROSS-DOMAIN POLICY ADAPTATION UNDER LIMITED SAMPLES

In this section, we start by theoretically unpacking the performance difference between a policy in the true target domain and the empirical source domain. Motivated by theoretical insights, we formulate a novel objective for cross-domain offline RL with optimal transport and support constraints. Moreover, we introduce our practical algorithm to fulfill dynamics-aware offline policy adaptation.

### 3.1 THEORETICAL ANALYSIS GIVEN TWO OFFLINE DATASETS

Since we involve source domain offline data to facilitate policy learning in the target domain, it is vital to investigate how source domain data can affect the performance of the learned policy in the target domain. To that end, we derive the performance bound of any policy $\pi$ in the target domain and the empirical source domain below, where $D_{\text{TV}}(p||q)$ denotes the total variation deviation between the probability distributions $p, q$. Due to space limits, all proofs are deferred to Appendix B.

**Theorem 3.1.** *Denote the empirical policy distribution in the offline dataset $D_{\text{src}}$ from source domain $\mathcal{M}_{\text{src}}$ and the offline dataset $D_{\text{tar}}$ from target domain $\mathcal{M}_{\text{tar}}$ as $\pi_{D_{\text{src}}} := \frac{\sum_{D_{\text{src}}} \mathbb{1}(s,a)}{\sum_{D_{\text{src}}} \mathbb{1}(s)}$ and $\pi_{D_{\text{tar}}} := \frac{\sum_{D_{\text{tar}}} \mathbb{1}(s,a)}{\sum_{D_{\text{tar}}} \mathbb{1}(s)}$, respectively. Denote $C_1 = \frac{2r_{\max}}{(1-\gamma)^2}$, then the return difference of any policy $\pi$ between the empirical source domain $\widehat{\mathcal{M}}_{\text{src}}$ and the true target domain $\mathcal{M}_{\text{tar}}$ is bounded:*

$$J_{\mathcal{M}_{\text{tar}}}(\pi) - J_{\widehat{\mathcal{M}}_{\text{src}}}(\pi) \geq -C_1 \underbrace{\mathbb{E}_{\rho^{\pi_{D_{\text{src}}}}_{\widehat{\mathcal{M}}_{\text{src}}}, P_{\widehat{\mathcal{M}}_{\text{src}}}}[D_{\text{TV}}(\pi_{D_{\text{src}}}||\pi)]}_{(a):\text{ source policy deviation}} - C_1 \underbrace{\mathbb{E}_{\rho^{\pi}_{\mathcal{M}_{\text{tar}}}, P_{\mathcal{M}_{\text{tar}}}}[D_{\text{TV}}(\pi_{D_{\text{tar}}}||\pi)]}_{(b):\text{ target policy deviation}}$$

$$- C_1 \underbrace{\mathbb{E}_{\rho^{\pi_D}_{\mathcal{M}_{\text{src}}}, \pi_{D_{\text{src}}}}\left[D_{\text{TV}}(P_{\mathcal{M}_{\text{tar}}}||P_{\widehat{\mathcal{M}}_{\text{src}}})\right]}_{(c):\text{ dynamics mismatch}} - \text{constant}.$$

**Remark:** The above bound depicts that the performance deviation of a policy in the true target domain and the empirical source domain is determined by three factors: term (a) and (b) that measure the deviation between the learned policy and the behavior policy in the source domain dataset and the target domain dataset, respectively; term (c) that measures the dynamics mismatch between the target domain dynamics and the empirical transition dynamics in the source domain dataset.

### 3.2 A NOVEL OBJECTIVE

Theorem 3.1 conveys that a promising way to improve the performance of the learned policy in the target domain when leveraging source domain offline data is to minimize the dynamics mismatch

between the target domain and the empirical source domain, *i.e.*, $D_{\mathrm{TV}}(P_{\mathcal{M}_{\mathrm{tar}}}\|P_{\widehat{\mathcal{M}}_{\mathrm{src}}})$. Naturally, this can be achieved by only keeping source domain data that have similar transition dynamics as the target domain dynamics. Motivated by this insight, we can formulate the objective function for training the value function $Q_\theta(s, a)$ parameterized by $\theta$ as:

$$\mathcal{L}_Q = \mathbb{E}_{D_{\mathrm{tar}}}\left[(Q_\theta - \mathcal{T}Q_\theta)^2\right] + \mathbb{E}_{(s,a,s')\sim D_{\mathrm{src}}}\left[\mathbb{1}(\hat{p} > \epsilon)(Q_\theta - \mathcal{T}Q_\theta)^2\right] + \mathcal{R}_{D_{\mathrm{src}}}(Q_\theta, \mathcal{T}Q_\theta), \quad (2)$$

where $\mathbb{1}(\cdot)$ is the indicator function, $\hat{p}$ measures the probability that the sampled source domain transition lies within the span of the target domain dataset, $\epsilon$ is a threshold hyperparameter, $\mathcal{T}$ is the Bellman operator, and $\mathcal{R}_{D_{\mathrm{src}}}(Q_\theta, \mathcal{T}Q_\theta)$ denotes the regularization term on source domain data. In practice, directly optimizing Equation 2 is not preferred since one may need to manually determine $\epsilon$ and adjust the regularization term per dataset. Instead, we propose to reject a fixed proposition of data in the sampled batch and combine the regularization term with the Bellman error to attain an *implicit* regularization. Formally, the objective function for the value function can be formulated as:

$$\mathcal{L}_Q = \mathbb{E}_{D_{\mathrm{tar}}}\left[(Q_\theta - \mathcal{T}Q_\theta)^2\right] + \mathbb{E}_{(s,a,s')\sim D_{\mathrm{src}}}\left[\omega(s, a, s')\mathbb{1}(\hat{p} > \hat{p}_{\xi\%})(Q_\theta - \mathcal{T}Q_\theta)^2\right], \quad (3)$$

where $\hat{p}_{\xi\%}$ is the top $\xi$-quantile likelihood estimation of the sampled batch from the source domain dataset, $\omega(s, a, s')$ is the weight that is related to $\hat{p}$, *i.e.*, $\omega(s, a, s')$ is large when $\hat{p}(s, a, s')$ is large (*i.e.*, $(s, a, s')$ lies close to the target domain) and vice versa. Equation 3 is appealing because it filters source domain data that deviate far from the target domain and adaptively penalizes the rest samples. Then, it remains to decide how to empirically calculate $\hat{p}$ and $\omega(s, a, s')$. We notice that measuring $\hat{p}$ can be equivalent to measuring the deviation of the source domain sample against the target domain. Some studies estimate the dynamics gap by training domain classifiers (Liu et al., 2022a), performing contrastive learning (Wen et al., 2024), etc. However, they often involve training neural networks, which can be inferior given few target domain data due to overfitting. Instead, we propose to adopt the *optimal transport* (OT) approach, which is a principled method for comparing two distributions, to align data in the source domain dataset and the target domain dataset.

We define $u = s_{\mathrm{src}} \oplus a_{\mathrm{src}} \oplus s'_{\mathrm{src}}$ and $u' = s_{\mathrm{tar}} \oplus a_{\mathrm{tar}} \oplus s'_{\mathrm{tar}}$, where $\oplus$ is the the vector concatenation operator, $(s_{\mathrm{src}}, a_{\mathrm{src}}, s'_{\mathrm{src}}) \sim D_{\mathrm{src}}, (s_{\mathrm{tar}}, a_{\mathrm{tar}}, s'_{\mathrm{tar}}) \sim D_{\mathrm{tar}}$. Let $p_s = \frac{1}{|D_{\mathrm{src}}|}\sum_{t=1}^{|D_{\mathrm{src}}|}\delta_{u_t}$ and $p_t = \frac{1}{|D_{\mathrm{tar}}|}\sum_{t=1}^{|D_{\mathrm{tar}}|}\delta_{u'_t}$ denote the state-action-next-state joint distribution of the source domain dataset and the target domain dataset, respectively. Given a cost function $C$, the Wasserstein distance

$$\mathcal{W}(u, u') = \min_{\mu \in M} \sum_{t=1}^{|D_{\mathrm{src}}|} \sum_{t'=1}^{|D_{\mathrm{tar}}|} C(u_t, u'_{t'})\mu_{t,t'} \quad (4)$$

can depict the distance between datasets from two domains. Suppose the optimal coupling by solving the optimization problem in Equation 4 gives $\mu^*$, we determine the deviation between a source domain data and the target domain dataset via:

$$d(u_t) = -\sum_{t'=1}^{|D_{\mathrm{tar}}|} C(u_t, u'_{t'})\mu^*_{t,t'}, \quad u_t = (s^t_{\mathrm{src}}, a^t_{\mathrm{src}}, (s'_{\mathrm{src}})^t) \sim D_{\mathrm{src}}. \quad (5)$$

Intuitively, $d$ is larger if the source domain data aligns the distribution of the target domain dataset (since the cost is smaller by then) and smaller otherwise. It can hence work as a good proxy for $\hat{p}$ in Equation 3. Furthermore, we show in Theorem 3.2 that $d$ serves as an upper bound of the negative total variation deviation between two empirical distributions $p_s, p_t$.

**Theorem 3.2.** *Assume that the cost is bounded, i.e.,* $C(u, u') \le C_{\max} < \infty, \forall\, u, u'$, *then we have*

$$0 \ge d(u_t) \ge -C_{\max}D_{\mathrm{TV}}(p_s\|p_t).$$

The assumption can be easily satisfied with an appropriate cost function (*e.g.*, Euclidean distance) since the state space and action space are usually bounded. The above theorem further validates the rationality of using $d$ where it bounds the dynamics discrepancy between two offline datasets.

Another benefit of computing $d$ is that it provides a natural way of measuring the weight $\omega(s, a, s')$ in Equation 3, *e.g.*, $\omega(s, a, s') = \exp(\alpha \times d(u))$, where $\alpha > 0$ is a hyperparameter. This is valid because $\omega \in [0, 1]$ given $d(u) \le 0$. Consequently, the objective function becomes

$$\mathcal{L}_Q = \mathbb{E}_{D_{\mathrm{tar}}}\left[(Q_\theta - \mathcal{T}Q_\theta)^2\right] + \mathbb{E}_{(s,a,s')\sim D_{\mathrm{src}}}\left[\exp(\alpha \times d)\mathbb{1}(d > d_{\xi\%})(Q_\theta - \mathcal{T}Q_\theta)^2\right], \quad (6)$$

where we abbreviate $d(u) = d(s, a, s')$ as $d$ and $d_{\xi\%}$ denotes the top $\xi$-quantile likelihood estimation of the sampled source domain batch. However, it is insufficient to merely optimize Equation 6 since the performance bound in Theorem 3.1 is also connected with policy deviation terms. If only limited target domain data is available (*e.g.*, 5000 transitions), the learned policy can get *biased* towards the behavior policy of the source domain dataset, incurring unsatisfying policy adaptation performance.

As a remedy, we include an extra policy regularization term that encourages the learned policy to be close to the support region of the target domain dataset (namely, *dataset regularization*). Let $\mathcal{L}_\pi$ be the vanilla policy optimization objective of the underlying offline RL algorithm, we modify it into

$$\widehat{\mathcal{L}}_\pi = \mathcal{L}_\pi - \beta \times \mathbb{E}_{s \sim D_{\mathrm{src}} \cup D_{\mathrm{tar}}} \log \pi_{\mathrm{tar}}^b(\pi(\cdot|s)|s), \tag{7}$$

where $\beta > 0$ is the weight coefficient, $\pi_{\mathrm{tar}}^b$ is the behavior policy in the target domain dataset. Then, minimizing $\widehat{\mathcal{L}}_\pi$ guarantees that the probability of the learned policy lying in the span of the target domain dataset is maximized. In this way, term (b) in Theorem 3.1 can be better controlled. Term (a) can also be minimized since the agent trains upon the source domain data.

Formally, we introduce our novel Optimal Transport Data Filtering (tagged OTDF) approach with its framework outlined in Figure 1, which is built upon the objective functions in Equation 6 and Equation 7. In principle, our framework can be integrated into any off-the-shelf offline RL algorithms. The abstracted pseudocode of OTDF is presented in Algorithm 1.

---

**Algorithm 1** Optimal Transport Data Filtering (Abstracted Version)

---

**Input:** Source domain dataset $D_{\mathrm{src}}$, target domain dataset $D_{\mathrm{tar}}$, batch size $N$, data selection ratio $\xi$
  1: Initialize policy $\pi_\phi$, value function $Q_\theta$, the cost function $C$, coefficients $\alpha, \beta$
  2: Compute the optimal alignment between $D_{\mathrm{src}}$ and $D_{\mathrm{tar}}$ with Equation 4        // Solve OT
  3: Compute deviations $\{d_t\}_{t=1}^{|D_{\mathrm{src}}|}$ between the source domain data and $D_{\mathrm{tar}}$ with Equation 5
  4: Concatenate $D_{\mathrm{src}}$ and $\{d_t\}_{t=1}^{|D_{\mathrm{src}}|}$ to get $\widehat{D}_{\mathrm{src}} = \{(s_t, a_t, r_t, s_t', d_t))\}_{t=1}^{|D_{\mathrm{src}}|}$
  5: **for** $i = 1, 2, ...$ **do**
  6:     Sample a mini-batch $b_{\mathrm{src}} := \{(s, a, r, s', d)\}$ with size $\frac{N}{2}$ from $D_{\mathrm{src}}$
  7:     Sample a mini-batch $b_{\mathrm{tar}} := \{(s, a, r, s')\}$ with size $\frac{N}{2}$ from $D_{\mathrm{tar}}$
  8:     Optimize the value function $Q_\theta$ on $b_{\mathrm{src}} \cup b_{\mathrm{tar}}$ with Equation 6        // Data filtering
  9:     Optimize the policy $\pi_\phi$ on $b_{\mathrm{src}} \cup b_{\mathrm{tar}}$ with Equation 7        // Dataset regularization
10: **end for**

---

### 3.3 Practical Algorithm

Intuitively, OTDF involves two crucial novel components: (a) selective source domain data sharing via optimal transport alignment, and (b) policy constraint to align with the target domain dataset.

**For the first component**, one can solve the OT problem in Equation 4 before initializing the offline RL algorithm, or during the policy optimization iterations. In practice, we resort to the former to save time used in solving the repetitive OT problem under different seeds, as shown in Algorithm 1. To obtain the optimal coupling matrix $\mu^*$ in Equation 4, we solve the entropy-regularized OT problem with Sinkhorn's algorithm (Cuturi, 2013), using the Sinkhorn solver in OTT-JAX (Cuturi et al., 2022). The OTT-JAX library provides a highly efficient and scalable implementation of the Sinkhorn algorithm using GPUs. As a result, we can compute the deviations in Equation 5 for 1 million source domain transitions within five minutes, given about 5000 target domain data.

We then can leverage Equation 6 to train the value function. However, the resulting deviations $d$ can be largely affected by the underlying task (*e.g.*, different tasks have varied state spaces and action spaces) and dataset qualities (*e.g.*, $d$ obtained from the medium-level source domain dataset and the expert-level source domain dataset can differ, given the same target domain dataset). This indicates that one may manually decide the coefficient $\alpha$ per dataset to acquire suitable weights, which hinders the practical application of OTDF. To mitigate this concern, we propose to *normalize* the deviations:

$$\widehat{d_i} = \frac{d_i - \max_i d_i}{\max_i d_i - \min_i d_i}, \quad i \in \{1, 2, \ldots, N\}, \tag{8}$$

where $N$ is the size of the sampled source domain batch. We subtract $\max_i d_i$ to ensure that $\widehat{d_i}$ lies in the range of $[-1, 0]$. We find this min-max normalization approach quite effective, diminishing

the need for the extra hyperparameter $\alpha$. We therefore optimize the following objective alternatively:

$$\mathcal{L}_Q = \mathbb{E}_{D_{\text{tar}}} \left[ (Q_\theta - \mathcal{T}Q_\theta)^2 \right] + \mathbb{E}_{(s,a,s') \sim D_{\text{src}}} \left[ \exp(\widehat{d}) \mathbb{1}(d > d_{\xi\%})(Q_\theta - \mathcal{T}Q_\theta)^2 \right]. \quad (9)$$

**For the second component**, it necessities to model the behavior policy of the target domain dataset. We fulfill this by using the conditional variational auto-encoder (CVAE) (Kingma & Welling, 2013), which we find can model the behavior policy well even under a limited budget of data. The CVAE $G$ consists of an encoder $E_\nu(s, a)$ and a decoder $D_\varsigma(s, z)$ parameterized by $\nu, \varsigma$ respectively. The objective function for training the CVAE in the target domain dataset gives,

$$\mathcal{L}_{\text{CVAE}} = \mathbb{E}_{(s,a) \sim D_{\text{tar}}, z \sim E_\nu(s,a)} \left[ (a - D_\varsigma(s, z))^2 + D_{\text{KL}} \left( E_\nu(s, a) \| \mathcal{N}(0, \mathbf{I}) \right) \right], \quad (10)$$

where $D_{\text{KL}}(p\|q)$ denotes the KL-divergence between two probability distributions $p, q$, and $\mathbf{I}$ is the identity matrix. We pretrain the CVAE policy upon the target domain dataset before training OTDF. Given a state sampled from the mixed dataset $D_{\text{src}} \cup D_{\text{tar}}$, we sample the corresponding action from the learned policy $a \sim \pi(\cdot|s)$, and feed them into the encoder $E$ to produce $M$ latent variables $z$. Afterward, we input $s$ and $z$ into the decoder $D$ to reconstruct actions $\widehat{a}$ that come from the same distribution as the target domain dataset and construct Gaussian distributions based on them. We then compute the log probability of $a$ (from $\pi$) belonging to these Gaussian distributions. Finally, we approximate Equation 7 by measuring the `logsumexp` of the log probabilities:

$$\widehat{\mathcal{L}}_\pi = \mathcal{L}_\pi - \beta \times \mathbb{E}_{s \sim D_{\text{src}} \cup D_{\text{tar}}} \log \left[ \sum_{i=1}^M \exp(\log \widehat{\pi}_{\text{tar}}^i(\pi(\cdot|s)|s)) \right], \quad (11)$$

where $\widehat{\pi}_{\text{tar}}^i$ denotes the $i$-th constructed Gaussian distribution with the decoded $\widehat{a}_i$ (based on $z_i, i \in \{1, \dots, M\}$) as the mean and a constant as the standard deviation. We find that OTDF is robust to different choices of $M$. We hence set $M = 10$ by default and do not bother tuning it. Apparently, Equation 9 and Equation 11 do not alter the vanilla objectives of the base offline RL methods and can serve as an add-on module for them. In this work, we build the practical OTDF algorithm upon IQL (Kostrikov et al., 2022) and defer the detailed pseudocode of OTDF+IQL to Appendix D.2.

## 4 EXPERIMENTS

In this section, we examine the effectiveness of our proposed method by conducting experiments on environments with various dynamics shifts. We aim to answer two questions: (a) Can OTDF fulfill effective offline policy adaptation and boost the performance of the base method? (b) Can OTDF beat prior strong baselines across varied dynamics shifts and dataset qualities? We further present a detailed parameter study to promote a better understanding of OTDF.

### 4.1 MAIN RESULTS

**Tasks and datasets.** To comprehensively evaluate the policy adaptation capabilities, we consider three kinds of dynamics shifts, including gravity shift, kinematic shift, and morphology shift, for four environments (*halfcheetah*, *hopper*, *walker2d*, *ant*) from OpenAI Gym (Brockman et al., 2016). The gravity shifts are realized by modifying the strength of the gravity (the direction of the gravitational force remains unchanged). We simulate the kinematic shifts by limiting the rotation range of some joints, and the morphology shifts by modifying the size of some limbs. More details on the environment settings can be found in Appendix C. Since we consider the cross-domain RL setting where only limited target domain data can be accessed, we use D4RL (Fu et al., 2020) "-v2" MuJoCo datasets as the *source domain datasets* and manually gather offline datasets in those modified environments to serve as *target domain datasets*, which contain only around 5000 transitions. This poses considerable challenges for existing offline RL methods to achieve good performance when merely training on the target domain, as observed in Liu et al. (2024a); Wen et al. (2024). We follow a similar data collection procedure as D4RL to collect *medium*, *medium-expert*, and *expert* target domain datasets. We allow source domain dataset qualities to be *medium*, *medium-replay* and *medium-expert*. This amounts to a total of **36** tasks for a single shift.

**Metrics.** We are primarily interested in the performance of the agent in the target domain. Since the scales of the return can differ in varied environments, we follow D4RL and evaluate the *normalized*

*score* metric: $NS = \frac{J-J_r}{J_e-J_r} \times 100$, where $J, J_e, J_r$ denote the return of the leaned policy, the expert policy and the random policy, respectively. $NS = 100$ corresponds to an expert policy and $NS = 0$ corresponds to a random policy. Please see Appendix C.1 for more details on the datasets.

**Baselines.** We consider the following baselines: **IQL\*** (Kostrikov et al., 2022) [1] that train the IQL policy upon both source domain data and target domain data; **DARA** (Liu et al., 2022a) that estimates the dynamics gap by training domain classifiers and utilizes it for penalizing source domain rewards; **BOSA** (Liu et al., 2024a) that employs support-constrained objectives to regularize the value function and the policy from being OOD; **SRPO** (Xue et al., 2024) that leverages the stationary state distribution as a regularizer for reward modification; **IGDF** (Wen et al., 2024) that filters source domain data by introducing a contrastive learning objective. The implementation details of these baselines can be found in Appendix D.1.

**Results.** We run all algorithms for 1M gradient steps across 5 random seeds. We summarize performance comparison results of OTDF against baselines under *morphology shifts* and *gravity shifts* in Table 1 and Table 2, respectively. Due to space limits, the empirical results under the *kinematic shifts* are deferred to Appendix E.1. We report the normalized score performance in the target domain.

*Answering question (a):* Based on the empirical results, it is evident that OTDF significantly outperforms IQL\* in most scenarios. Notably, OTDF achieves higher normalized scores than IQL\* on **27** out of 36 tasks under the morphology shifts and **29** out of 36 tasks under the gravity shifts. OTDF exhibits competitive performance as IQL\* on the remaining tasks. Adopting OTDF incurs an increase of **59.7%** and **40.7%** in terms of the total normalized score under the morphology and gravity shift tasks, respectively. These clearly validate the effectiveness of our method.

*Answering question (b):* We find that OTDF significantly surpasses recent baselines across numerous dataset qualities and types of the dynamics shift scenarios, often by a large margin. Specifically, OTDF excels in **23** out of 36 tasks (with varied dataset qualities of both domains) under the morphology shifts and achieves a total normalized score **1274.3**, while the second best baseline (DARA) only has a total score of 816.8. Under the gravity shift tasks, OTDF markedly beats other methods across **26** out of 36 tasks, exceeding the second best approach (DARA) by **36.4%** in terms of the total normalized score performance. Despite OTDF's suboptimal performance in some tasks, it still remains competitive compared to other methods in those tasks.

We observe that the most recent baselines have similar performance as IQL\* on many tasks, indicating that they fail to fulfill effective offline policy adaptation. *OTDF is the only method that exhibits remarkable performance gain compared to the base method.* This can be possibly attributed to the fact that it is difficult to learn classifiers (DARA, SRPO) or dynamics transition model (BOSA) given a limited budget of the target domain dataset. The reward penalties provided by DARA and SRPO can hence be poor, which ultimately results in negative effects on policy training. Furthermore, the policy trained by BOSA can be *biased* and favor the distribution of the source domain dataset instead of the target domain dataset, because BOSA constrains the learned policy to stay within the support region of the *mixed dataset* $D_{\mathrm{src}} \cup D_{\mathrm{tar}}$. OTDF lifts these concerns by selectively sharing source domain data with OT (no neural network training) and enforcing target domain dataset regularization. Since the computation of OT is not affected by the coverage or the quality of the adopted datasets, OTDF can consistently keep its advantages over baselines across numerous tasks regardless of the dataset qualities from both domains and types of dynamics shifts.

## 4.2 Parameter Study

In this part, we investigate how sensitive OTDF is to the introduced hyperparameters. There are two crucial hyperparameters in OTDF, the data selection ratio $\xi$ and the policy coefficient $\beta$.

**Data selection ratio $\xi$.** $\xi$ decides how many source domain data in a sampled batch can be shared for policy training. A larger $\xi$ indicates that more source domain data will be admitted. To examine its influence, we conduct experiments with *medium* source domain datasets and *medium* target domain datasets and sweep $\xi$ across $\{0, 20, 40, 60, 80, 100\}$. $\xi = 100$ means that no data selection process is included in OTDF while $\xi = 0$ means that all source domain data are rejected when learning the value function. We summarize the experimental results in Figure 2 and observe that it is not ideal to set $\xi = 0$ or 100. This ablates the necessity of the data filtering process with OT. Note that different

---

[1] We add a \* to IQL to highlight that it is trained on the mixed dataset.

Table 1: **Performance comparison in cross-domain offline RL given the *morphology* shift**. half = halfcheetah, hopp = hopper, walk = walker2d, m = medium, r = replay, e = expert. The target column denotes the offline dataset quality of the target domain. We report normalized scores and standard deviations in the *target domain* under varied dataset qualities of the source domain data (medium, medium-replay, medium-expert) and target domain data (medium, medium-expert, expert). The results are averaged over 5 seeds. We **bold** and highlight the best cell.

| Source | Target | IQL* | DARA | BOSA | SRPO | IGDF | OTDF (ours) |
|---|---|---|---|---|---|---|---|
| half-m | medium | 30.0±1.6 | 26.6±3.3 | 19.3±3.5 | 41.3±0.4 | **41.6**±0.5 | 39.1±2.3 |
| half-m | medium-expert | 31.8±1.1 | 32.0±0.7 | 33.6±1.1 | 30.7±0.8 | 29.6±2.2 | **35.6**±0.7 |
| half-m | expert | 8.5±1.0 | 9.3±1.6 | 7.9±0.8 | 8.6±0.9 | 10.0±0.8 | **10.7**±1.2 |
| half-m-r | medium | 30.8±4.4 | 35.6±0.7 | 35.0±4.6 | 32.0±1.4 | 28.0±2.0 | **40.0**±1.2 |
| half-m-r | medium-expert | 12.9±2.2 | 16.9±4.1 | 19.9±5.5 | 12.4±1.6 | 12.0±3.7 | **34.4**±0.7 |
| half-m-r | expert | 5.9±1.7 | 3.7±2.7 | 2.4±1.9 | 6.2±1.4 | 5.3±2.3 | **8.2**±2.7 |
| half-m-e | medium | **41.5**±0.1 | 40.3±1.2 | 41.3±0.3 | 41.3±0.4 | 40.9±0.4 | 41.4±0.3 |
| half-m-e | medium-expert | 25.8±2.0 | 30.6±2.8 | 32.1±0.8 | 27.2±0.8 | 26.2±1.8 | **35.1**±0.6 |
| half-m-e | expert | 7.8±1.3 | 8.3±1.3 | 9.1±0.8 | 7.8±0.9 | 7.5±0.9 | **9.8**±1.0 |
| hopp-m | medium | **13.5**±0.2 | **13.5**±0.4 | 13.2±0.3 | 13.4±0.1 | 13.4±0.2 | 11.0±0.9 |
| hopp-m | medium-expert | 13.4±0.1 | **13.6**±0.2 | 11.2±4.6 | 13.3±0.2 | 13.3±0.4 | 12.6±0.8 |
| hopp-m | expert | 13.5±0.2 | 13.6±0.3 | 13.3±0.4 | 13.6±0.2 | **13.9**±0.1 | 10.7±4.7 |
| hopp-m-r | medium | 10.8±1.1 | 10.2±1.0 | 1.2±0.0 | 10.7±1.6 | **12.0**±4.4 | 8.7±2.8 |
| hopp-m-r | medium-expert | **11.6**±1.6 | 10.4±0.9 | 1.3±0.2 | 10.4±1.2 | 8.2±2.8 | 9.7±2.7 |
| hopp-m-r | expert | 9.8±0.5 | 9.0±0.3 | 1.3±0.1 | 10.4±1.4 | **11.4**±1.5 | 10.7±2.4 |
| hopp-m-e | medium | 12.6±1.4 | 13.0±0.5 | **15.7**±7.2 | 14.0±2.3 | 12.7±0.8 | 7.9±3.2 |
| hopp-m-e | medium-expert | **14.1**±1.3 | 13.8±0.6 | 12.0±1.4 | 13.5±0.3 | 13.3±1.2 | 9.6±3.5 |
| hopp-m-e | expert | 13.8±0.5 | 12.3±1.8 | 10.5±5.0 | **14.7**±2.3 | 12.8±0.9 | 5.9±4.0 |
| walk-m | medium | 23.0±4.7 | 23.3±3.3 | 6.2±2.9 | 24.7±1.7 | 27.5±9.5 | **50.5**±5.8 |
| walk-m | medium-expert | 21.5±8.6 | 22.2±7.6 | 7.2±2.9 | 18.7±7.3 | 20.7±5.9 | **44.3**±23.8 |
| walk-m | expert | 20.3±2.8 | 17.3±3.4 | 15.8±8.7 | 21.1±7.2 | 15.8±4.5 | **55.3**±8.3 |
| walk-m-r | medium | 11.3±3.0 | 10.9±4.6 | 5.4±4.0 | 10.4±4.8 | 13.4±7.2 | **37.4**±5.1 |
| walk-m-r | medium-expert | 7.0±1.5 | 4.5±1.1 | 4.0±2.2 | 4.9±1.7 | 6.9±2.2 | **33.8**±6.9 |
| walk-m-r | expert | 6.3±0.9 | 4.5±1.1 | 3.8±3.4 | 5.5±0.9 | 5.5±2.2 | **41.5**±6.8 |
| walk-m-e | medium | 24.1±7.4 | 31.7±6.6 | 18.7±6.5 | 29.9±4.7 | 27.5±2.3 | **49.9**±4.6 |
| walk-m-e | medium-expert | 27.0±5.5 | 23.3±5.5 | 11.1±0.9 | 22.9±3.8 | 25.3±6.4 | **40.5**±11.0 |
| walk-m-e | expert | 22.4±3.3 | 25.2±5.7 | 9.9±3.9 | 18.7±5.7 | 24.7±2.4 | **45.7**±6.9 |
| ant-m | medium | 38.7±3.8 | **41.3**±1.8 | 18.2±1.9 | 40.6±2.1 | 40.9±1.7 | 39.4±1.7 |
| ant-m | medium-expert | 47.0±5.1 | 43.3±2.0 | 45.3±7.0 | 47.2±4.3 | 44.4±1.7 | **58.3**±8.9 |
| ant-m | expert | 36.2±3.5 | 48.5±4.2 | 72.2±10.5 | 42.2±9.9 | 41.4±4.2 | **85.4**±4.4 |
| ant-m-r | medium | 38.2±2.9 | 38.9±2.7 | 20.2±3.7 | 38.3±1.9 | 39.7±1.2 | **41.2**±0.9 |
| ant-m-r | medium-expert | 38.1±3.5 | 33.4±5.5 | 15.2±1.6 | 35.0±5.7 | 37.3±2.4 | **50.8**±4.5 |
| ant-m-r | expert | 24.1±1.9 | 24.5±2.6 | 16.0±1.7 | 22.7±3.0 | 23.6±1.4 | **67.2**±7.5 |
| ant-m-e | medium | 32.9±5.1 | **40.2**±1.5 | 28.1±5.6 | 35.9±2.5 | 36.1±4.4 | 39.9±2.9 |
| ant-m-e | medium-expert | 35.7±3.9 | 36.5±8.7 | 14.8±15.9 | 24.5±15.7 | 30.7±10.8 | **65.7**±4.5 |
| ant-m-e | expert | 36.1±8.5 | 34.6±5.8 | 53.9±5.0 | 38.4±9.4 | 35.2±6.6 | **86.4**±2.2 |
| | Total Score | 798.0 | 816.8 | 646.3 | 803.1 | 808.7 | **1274.3** |

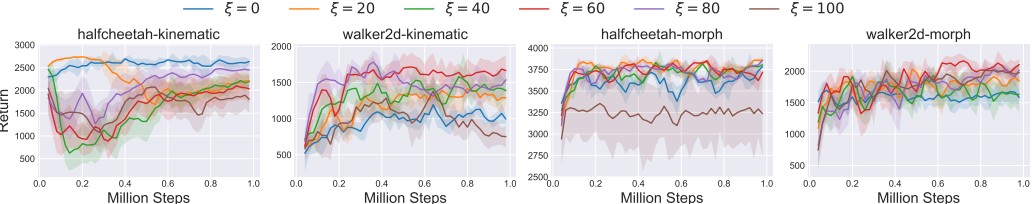

Figure 2: **Parameter study of the data selection ratio $\xi$.** *-kinematic denotes the kinematic shift task, while *-morph denotes the morphology shift tasks. We use *halfcheetah*, *walker2d* tasks. The solid lines depict the average returns over 5 seeds and the shaded area denotes the standard deviation.

Table 2: **Performance comparison under the *gravity* shift**. half = halfcheetah, hopp = hopper, walk = walker2d, m = medium, r = replay, e = expert. The target column denotes the offline dataset quality of the target domain. The normalized average scores in the target domain across 5 random seeds are reported and $\pm$ captures the standard deviation. We **bold** and highlight the best cell.

| Source | Target | IQL* | DARA | BOSA | SRPO | IGDF | OTDF (ours) |
|---|---|---|---|---|---|---|---|
| half-m | medium | 39.6±3.3 | **41.2**±3.9 | 38.9±4.0 | 36.9±4.5 | 36.6±5.5 | 40.7±7.7 |
| half-m | medium-expert | 39.6±3.7 | **40.7**±2.8 | 40.4±3.0 | 40.7±2.3 | 38.7±6.2 | 28.6±3.2 |
| half-m | expert | **42.4**±3.8 | 39.8±4.4 | 40.5±3.9 | 39.4±1.6 | 39.6±4.6 | 36.1±5.3 |
| half-m-r | medium | 20.1±5.0 | 17.6±6.2 | 20.0±4.9 | 17.5±5.2 | 14.4±2.2 | **21.5**±6.5 |
| half-m-r | medium-expert | 17.2±1.6 | **20.2**±5.2 | 16.7±4.2 | 16.3±1.7 | 10.0±2.5 | 14.7±4.1 |
| half-m-r | expert | 20.7±5.5 | 22.4±1.7 | 15.4±4.2 | **23.1**±4.0 | 15.3±3.7 | 11.4±1.9 |
| half-m-e | medium | 38.6±6.0 | 37.8±3.3 | 41.8±5.1 | **42.5**±2.3 | 37.7±7.3 | 39.5±3.5 |
| half-m-e | medium-expert | 39.6±3.0 | 39.4±4.4 | 38.7±3.7 | **43.3**±2.7 | 40.7±3.2 | 32.4±5.5 |
| half-m-e | expert | 43.4±0.9 | **45.3**±1.3 | 39.9±2.7 | 43.3±3.0 | 41.1±4.1 | 26.5±9.1 |
| hopp-m | medium | 11.2±1.1 | 17.3±3.8 | 15.2±3.3 | 12.4±1.0 | 15.3±3.5 | **32.4**±8.0 |
| hopp-m | medium-expert | 14.7±3.6 | 15.4±2.5 | 21.1±9.3 | 14.2±1.8 | 15.1±3.6 | **24.2**±3.6 |
| hopp-m | expert | 12.5±1.6 | 19.3±10.5 | 12.7±1.7 | 11.8±0.9 | 14.8±4.0 | **33.7**±7.8 |
| hopp-m-r | medium | 13.9±2.9 | 10.7±4.3 | 3.3±1.9 | 14.0±2.6 | 15.3±4.4 | **31.1**±13.4 |
| hopp-m-r | medium-expert | 13.3±6.3 | 12.5±5.6 | 4.6±1.7 | 14.4±4.2 | 15.4±5.5 | **24.2**±6.1 |
| hopp-m-r | expert | 11.0±2.6 | 14.3±6.0 | 3.2±0.8 | 16.4±5.0 | 16.1±4.0 | **31.0**±9.8 |
| hopp-m-e | medium | 19.1±6.6 | 18.5±12.3 | 15.9±5.9 | 19.7±8.5 | 22.3±5.4 | **26.4**±10.1 |
| hopp-m-e | medium-expert | 16.8±2.7 | 16.0±6.1 | 17.3±2.5 | 15.8±3.3 | 16.6±7.7 | **28.3**±6.7 |
| hopp-m-e | expert | 20.9±4.1 | 23.9±14.8 | 23.2±7.9 | 21.4±1.9 | 26.0±9.2 | **44.9**±10.6 |
| walk-m | medium | 28.1±12.9 | 28.4±13.7 | **38.0**±11.2 | 21.4±7.0 | 22.1±8.4 | 36.6±2.3 |
| walk-m | medium-expert | 35.7±4.7 | 30.7±9.7 | 40.9±7.2 | 34.0±9.9 | 35.4±9.1 | **44.8**±7.5 |
| walk-m | expert | 37.3±8.0 | 36.0±7.0 | 41.3±8.6 | 39.5±3.8 | 36.2±13.6 | **44.0**±4.0 |
| walk-m-r | medium | 14.6±2.5 | 14.1±6.1 | 7.6±5.8 | 17.9±3.8 | 11.6±4.6 | **32.7**±7.0 |
| walk-m-r | medium-expert | 15.3±1.9 | 15.9±5.8 | 4.8±5.8 | 15.3±4.5 | 13.9±6.5 | **31.6**±6.1 |
| walk-m-r | expert | 15.8±7.2 | 15.7±4.5 | 7.1±4.6 | 13.7±8.1 | 15.2±5.3 | **31.3**±5.3 |
| walk-m-e | medium | 39.9±13.1 | 41.6±13.0 | 32.3±7.2 | **46.4**±3.5 | 33.8±3.1 | 30.2±9.8 |
| walk-m-e | medium-expert | 49.1±6.9 | 45.8±9.4 | 40.1±4.5 | 36.4±3.4 | 44.7±2.9 | **53.3**±7.1 |
| walk-m-e | expert | 40.4±11.9 | 56.4±3.5 | 43.7±4.4 | 45.8±8.0 | 45.3±10.4 | **61.1**±3.4 |
| ant-m | medium | 10.2±1.8 | 9.4±0.9 | 12.4±2.0 | 11.7±1.0 | 11.3±1.3 | **45.1**±12.4 |
| ant-m | medium-expert | 9.4±1.2 | 10.0±0.9 | 11.6±1.3 | 10.2±1.2 | 9.4±1.4 | **33.9**±5.4 |
| ant-m | expert | 10.2±0.3 | 9.8±0.6 | 11.8±0.4 | 9.5±0.6 | 9.7±1.6 | **33.2**±9.0 |
| ant-m-r | medium | 18.9±2.6 | 21.7±2.1 | 13.9±1.5 | 18.7±1.7 | 19.6±1.0 | **29.6**±10.7 |
| ant-m-r | medium-expert | 19.1±3.0 | 18.3±2.1 | 15.9±2.7 | 18.7±1.8 | 20.3±1.6 | **25.4**±2.1 |
| ant-m-r | expert | 18.5±0.9 | 20.0±1.3 | 14.5±1.7 | 19.9±2.1 | 18.8±2.1 | **24.5**±2.8 |
| ant-m-e | medium | 9.8±2.4 | 8.1±1.8 | 8.1±3.0 | 8.4±2.1 | 8.9±1.5 | **18.6**±11.9 |
| ant-m-e | medium-expert | 9.0±0.8 | 6.4±1.4 | 6.2±1.5 | 6.1±3.5 | 7.2±2.9 | **34.0**±9.4 |
| ant-m-e | expert | 9.1±2.6 | 10.4±2.9 | 4.2±3.9 | 8.8±1.0 | 9.2±1.5 | **23.2**±2.9 |
| | Total Score | 825.0 | 851.0 | 763.2 | 825.5 | 803.6 | **1160.7** |

tasks prefer varied $\xi$ while we can achieve a trade-off with $\xi = 80$. We hence set $\xi = 80$ (*i.e.*, share 80% source domain data) for OTDF by default and do not tune it.

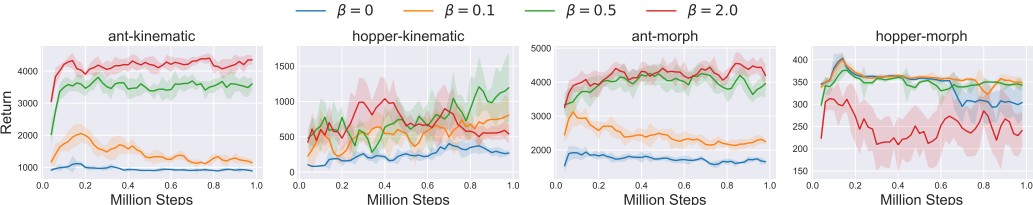

Figure 3: **Parameter study of the policy regularization coefficient $\beta$.** We report the returns obtained in the modified *hopper* and *ant* tasks. The shaded region captures the standard deviation.

**Policy coefficient $\beta$.** $\beta$ controls the strengths of the target domain constraint term in Equation 11. A larger $\beta$ may de-emphasize the knowledge carried by the source domain data while a small $\beta$ can

incur a biased policy that leans towards the distribution of the source domain dataset. Obviously, $\beta$ needs to be set properly to achieve a trade-off. We employ $\beta \in \{0, 0.1, 0.5, 2.0\}$ and run OTDF on the *medium-replay* source domain datasets and *expert* target domain datasets. We present the results in Figure 3 and find that diminishing the role of dataset regularization (*i.e.*, $\beta = 0$) leads to unsatisfying performance. OTDF can be sensitive to the choice of $\beta$ and the best $\beta$ for each task vary (*e.g.*, ant-kinematic prefers $\beta = 2.0$ while hopper-kinematic favors $\beta = 0.5$). Fortunately, we can reach a trade-off with $\beta = 0.5$ or $\beta = 0.1$. For most of our experiments, we set $\beta = 0.5$.

## 5    RELATED WORK

**Offline Reinforcement Learning (RL).** In Offline RL (or batch RL) (Levine et al., 2020; Lange et al., 2012), the agent is only allowed to learn policies based on some previously gathered offline datasets. Offline RL methods can be generally divided into model-free (Wu et al., 2021; An et al., 2021) and model-based (Argenson & Dulac-Arnold, 2020; Matsushima et al., 2021) approaches. The success of these offline RL methods relies heavily on the fact that the underlying static datasets usually contain a large amount of transitions. Instead, we investigate how to facilitate target domain policy training by leveraging source domain datasets, given only *few* target domain data.

**Domain Adaptation in RL.** In this work, we focus on the policy adaptation problem (Xu et al., 2023; Lyu et al., 2024a;b) under dynamics shifts between the two domains, while other components like observation spaces are unchanged. To mitigate this issue, previous methods mainly leverages domain randomization (Slaoui et al., 2019; Mehta et al., 2019), system identification (Clavera et al., 2018; Du et al., 2021), imitation learning (Kim et al., 2019; Hejna et al., 2020), and meta-RL methods (Nagabandi et al., 2018; Raileanu et al., 2020), etc. However, they often require expert demonstrations from the target domain, and prior knowledge to guide parameter randomization. Some recent works discard these demands and train dynamics-aware policies given limited offline transitions from one domain while having access to another online domain with dynamics discrepancies (Niu et al., 2022; Xu et al., 2023; Lyu et al., 2024a). In contrast, we explore the *offline policy adaptation* setting where the two domains are both offline. Existing works address this issue via performing reward penalization (Liu et al., 2022a), conducting data filtering from the perspective of mutual information (Wen et al., 2024), etc. Different from these methods, we resort to the *optimal transport* approach for selecting source domain transitions that are close to the target domain, in conjunction with a *policy constraint term* that enforces the learned policy to stay close to the support region of the target domain. These allow our method to function well even under very few target domain samples where existing methods typically fail.

**Optimal Transport (OT).** OT is widely used in domain adaptation (Damodaran et al., 2018; Shen et al., 2018), recommender system (Li et al., 2019; Mashayekhi et al., 2023), and graph matching (Chen et al., 2020; Xu et al., 2019). In the context of RL, OT is used in fields like imitation learning (Haldar et al., 2023; Nguyen et al., 2021; Dadashi et al., 2020; Luo et al., 2023), curriculum RL (Klink et al., 2022; Huang et al., 2022), preference-based RL (Liu et al., 2024b), etc. We, instead, resort to OT for data filtering in the context of cross-domain offline RL.

## 6    CONCLUSION

In this paper, we study the cross-domain offline policy adaptation problem which seeks to enhance offline policy training on a limited target domain dataset by leveraging a source domain dataset with dynamics shifts. We theoretically characterize the performance bound of a policy under this setting, which further motivates us to perform data filtering upon source domain data with optimal transport and introduce a dataset regularization term to maximize the probability that the learned policy stays within the span of the target domain dataset. These give birth to the OTDF algorithm, which is compatible with any offline RL methods. Empirically, we combine OTDF with IQL and evaluate its performance upon datasets with distinct qualities and dynamics shift types. Experimental results demonstrate that OTDF significantly exceeds baselines on numerous tasks, often by a large margin.

The limitations of our work lie in that: (a) OTDF is inapplicable when the source domain and the target domain have varied state spaces or action spaces; (b) our experiments are only carried out in the simulated environments, and it remains to explore the effectiveness of OTDF in real-world scenarios. We plan to resolve these drawbacks in future work.

ACKNOWLEDGEMENTS

This work was supported by the STI 2030-Major Projects under Grant 2021ZD0201404 and NSFC under Grant 62450001. The authors would like to thank the anonymous reviewers for their valuable comments and advice.

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

## A    EXTENDED RELATED WORK

**Offline Reinforcement Learning (RL).** In Offline RL (or batch RL) (Levine et al., 2020; Lange et al., 2012), the agent is only allowed to learn policies based on some previously gathered offline datasets. Offline RL methods can be generally divided into model-free (Wu et al., 2021; An et al., 2021) and model-based (Argenson & Dulac-Arnold, 2020; Matsushima et al., 2021) approaches, where model-free offline RL algorithms often leverage value penalization (Kumar et al., 2020; Lyu et al., 2022c; Nikulin et al., 2023; Yang et al., 2024; Yeom et al., 2024), implicit or explicit policy constraints (Fujimoto et al., 2019; Fujimoto & Gu, 2021; Kostrikov et al., 2022; Wu et al., 2022a; Lyu et al., 2022a; Ran et al., 2023; Tarasov et al., 2024), while model-based offline RL methods typically rely on directly optimizing the policy with the learned model (Yu et al., 2020; Kidambi et al., 2020; Rigter et al., 2022; Guo et al., 2022; Sun et al., 2023; Qiao et al., 2024; Luo et al., 2024), or performing data augmentation for offline datasets (Yu et al., 2021; Lyu et al., 2022b; Zhang et al., 2023). The success of these offline RL methods relies heavily on the fact that the underlying static datasets usually contain a large amount of transitions. Instead, we investigate how to facilitate target domain policy training by leveraging source domain datasets, given only *few* target domain data.

**Domain Adaptation in RL.** It is challenging to generalize or transfer policies to another domain (Cobbe et al., 2019) in RL, where the two domains can differ in terms of agent embodiment (Liu et al., 2022b; Zhang et al., 2021b), observation or action spaces (Gamrian & Goldberg, 2018; Bousmalis et al., 2018; Zhang et al., 2021a; Ge et al., 2022), and the environmental dynamics (Eysenbach et al., 2021; Viano et al., 2020), etc. In this work, we focus on the policy adaptation problem (Xu et al., 2023; Lyu et al., 2024a;b) under dynamics shifts between the two domains, while other components like observation spaces remain unchanged. To mitigate this challenge, previous literature mainly leverages domain randomization (Slaoui et al., 2019; Mehta et al., 2019; Vuong et al., 2019; Jiang et al., 2023), system identification (Clavera et al., 2018; Du et al., 2021; Xie et al., 2022), imitation learning (Kim et al., 2019; Hejna et al., 2020; Fickinger et al., 2022; Raychaudhuri et al., 2021; Guo et al., 2024), and meta-RL methods (Nagabandi et al., 2018; Raileanu et al., 2020; Arndt et al., 2019; Wu et al., 2022b), etc. Nevertheless, these methods often require expert demonstrations from the target domain, and prior knowledge to guide parameter randomization. Some recent works discard these demands and train dynamics-aware policies given limited offline transitions from one domain while having access to another online domain with dynamics discrepancies (Eysenbach et al., 2021; Niu et al., 2022; Xu et al., 2023; Lyu et al., 2024a). In contrast, we explore the *offline policy adaptation* setting where the two domains are purely offline. Existing works address this issue via performing reward penalization (Liu et al., 2022a), training a GAN-style discriminator (Xue et al., 2024), incorporating pessimistic supported regularization (Liu et al., 2024a), utilizing the return-conditioned supervised learning (RCSL) approach (Wang et al., 2024), or conducting data filtering from the perspective of mutual information (Wen et al., 2024). Different from these methods, we resort to the *optimal transport* approach for selecting source domain transitions that are close to the target domain, in conjunction with a *policy constraint term* that enforces the learned policy to stay close to the support region of the target domain. These allow our method to function well even under very few target domain samples where existing methods typically fail.

## B    MISSING PROOFS

In this section, we present the proofs of the theoretical results that are omitted from the main text due to space limits. We note that $Q_{\mathcal{M}}^{\pi}(s, a)$ means the state-action value function upon the sample $(s, a)$ by following the policy $\pi$ in the MDP $\mathcal{M}$. Also, recall that $P_{\mathrm{src}} = P_{\mathcal{M}_{\mathrm{src}}}, P_{\mathrm{tar}} = P_{\mathcal{M}_{tar}}$.

### B.1    LEMMAS

**Lemma B.1** (Telescoping lemma). *Denote $\mathcal{M}_1 = (\mathcal{S}, \mathcal{A}, P_1, r, \gamma)$ and $\mathcal{M}_2 = (\mathcal{S}, \mathcal{A}, P_2, r, \gamma)$ as two MDPs that only differ in their transition dynamics. Suppose we have two policies $\pi_1, \pi_2$, we can reach the following conclusion:*

$$J_{\mathcal{M}_1}(\pi_1) - J_{\mathcal{M}_2}(\pi_2) = \frac{1}{1 - \gamma} \mathbb{E}_{\rho_{\mathcal{M}_1}^{\pi_1}(s,a)} \left[ \mathbb{E}_{s' \sim P_1, a' \sim \pi_1}[Q_{\mathcal{M}_2}^{\pi_2}(s', a')] - \mathbb{E}_{s' \sim P_2, a' \sim \pi_2}[Q_{\mathcal{M}_2}^{\pi_2}(s', a')] \right].$$

$$(12)$$

*Proof.* Please check the proof of Lemma C.2 in Xu et al. (2023). □

**Lemma B.2.** *Denote $\mathcal{M} = (\mathcal{S}, \mathcal{A}, P, r, \gamma)$ as the underlying MDP., The performance difference of the two policies $\pi_1, \pi_2$ in the MDP $\mathcal{M}$ gives:*

$$J_{\mathcal{M}}(\pi_1) - J_{\mathcal{M}}(\pi_2) = \frac{1}{1-\gamma} \mathbb{E}_{\rho_{\mathcal{M}}^{\pi_1}(s,a), s' \sim P} \left[ \mathbb{E}_{a' \sim \pi_1}[Q_{\mathcal{M}}^{\pi_2}(s', a')] - \mathbb{E}_{a' \sim \pi_2}[Q_{\mathcal{M}}^{\pi_2}(s', a')] \right]. \quad (13)$$

*Proof.* Please check the proof of Lemma B.3 in Lyu et al. (2024a). □

## B.2 PROOF OF THEOREM 3.1

**Theorem B.3** (Offline performance bound). *Denote the empirical policy distribution in the offline dataset $D_{\mathrm{src}}$ from source domain $\mathcal{M}_{\mathrm{src}}$ and the offline dataset $D_{\mathrm{tar}}$ from target domain $\mathcal{M}_{\mathrm{tar}}$ as $\pi_{D_{\mathrm{src}}} := \frac{\sum_{D_{\mathrm{src}}} \mathbb{1}(s,a)}{\sum_{D_{\mathrm{src}}} \mathbb{1}(s)}$ and $\pi_{D_{\mathrm{tar}}} := \frac{\sum_{D_{\mathrm{tar}}} \mathbb{1}(s,a)}{\sum_{D_{\mathrm{tar}}} \mathbb{1}(s)}$, respectively. Denote $C_1 = \frac{2r_{\max}}{(1-\gamma)^2}$, then the return difference of any policy $\pi$ between the empirical source domain $\widehat{\mathcal{M}}_{\mathrm{src}}$ and the true target domain $\mathcal{M}_{\mathrm{tar}}$ is bounded:*

$$J_{\mathcal{M}_{\mathrm{tar}}}(\pi) - J_{\widehat{\mathcal{M}}_{\mathrm{src}}}(\pi) \geq -C_1 \underbrace{\mathbb{E}_{\rho_{\widehat{\mathcal{M}}_{\mathrm{src}}}^{\pi_{D_{\mathrm{src}}}}, P_{\widehat{\mathcal{M}}_{\mathrm{src}}}} [D_{\mathrm{TV}}(\pi_{D_{\mathrm{src}}} \| \pi)]}_{(a):\,\text{source policy deviation}} - C_1 \underbrace{\mathbb{E}_{\rho_{\mathcal{M}_{\mathrm{tar}}}^{\pi}, P_{\mathcal{M}_{\mathrm{tar}}}} [D_{\mathrm{TV}}(\pi_{D_{\mathrm{tar}}} \| \pi)]}_{(b):\,\text{target policy deviation}}$$

$$- C_1 \underbrace{\mathbb{E}_{\rho_{\mathcal{M}_{\mathrm{src}}}^{\pi_D}, \pi_{D_{\mathrm{src}}}} \left[ D_{\mathrm{TV}}(P_{\mathcal{M}_{\mathrm{tar}}} \| P_{\widehat{\mathcal{M}}_{\mathrm{src}}}) \right]}_{(c):\,\text{dynamics mismatch}} - \text{constant}.$$

*Proof.* We establish the performance bound of the policy in the true target domain $\mathcal{M}_{\mathrm{tar}}$ and the empirical source domain $\widehat{\mathcal{M}}_{\mathrm{src}}$ by dividing the performance deviation into three parts,

$$J_{\mathcal{M}_{\mathrm{tar}}}(\pi) - J_{\widehat{\mathcal{M}}_{\mathrm{src}}}(\pi) = \underbrace{(J_{\mathcal{M}_{\mathrm{tar}}}(\pi) - J_{\mathcal{M}_{\mathrm{tar}}}(\pi_{D_{\mathrm{tar}}}))}_{(a)} + \underbrace{\left( J_{\mathcal{M}_{\mathrm{tar}}}(\pi_{D_{\mathrm{tar}}}) - J_{\widehat{\mathcal{M}}_{\mathrm{src}}}(\pi_{D_{\mathrm{src}}}) \right)}_{(b)}$$

$$+ \underbrace{\left( J_{\widehat{\mathcal{M}}_{\mathrm{src}}}(\pi_{D_{\mathrm{src}}}) - J_{\widehat{\mathcal{M}}_{\mathrm{src}}}(\pi) \right)}_{(c)}. \quad (14)$$

The term $(a)$ in the RHS measures the performance deviation of the learned policy and the behavior policy of target domain dataset under the target domain, term $(b)$ measures the performance of the behavior policy of the target domain dataset in the true target domain MDP against the performance of the behavior policy in the source domain dataset under the empirical source domain MDP, and term $(c)$ depicts the performance deviation of the learned policy and the behavior policy in the source domain under the empirical source domain MDP. We first bound term $(a)$. By using Lemma B.2, we have

$$(a) = J_{\mathcal{M}_{\mathrm{tar}}}(\pi) - J_{\mathcal{M}_{\mathrm{tar}}}(\pi_{D_{\mathrm{tar}}})$$

$$= \frac{1}{1-\gamma} \mathbb{E}_{\rho_{\mathcal{M}_{\mathrm{tar}}}^{\pi}(s,a), s' \sim P_{\mathcal{M}_{\mathrm{tar}}}(\cdot|s,a)} \left[ \mathbb{E}_{a' \sim \pi} \left[ Q_{\mathcal{M}_{\mathrm{tar}}}^{\pi_{D_{\mathrm{tar}}}}(s', a') \right] - \mathbb{E}_{a' \sim \pi_{D_{\mathrm{tar}}}} \left[ Q_{\mathcal{M}_{\mathrm{tar}}}^{\pi_{D_{\mathrm{tar}}}}(s', a') \right] \right]$$

$$\geq -\frac{1}{1-\gamma} \mathbb{E}_{\rho_{\mathcal{M}_{\mathrm{tar}}}^{\pi}(s,a), s' \sim P_{\mathcal{M}_{\mathrm{tar}}}(\cdot|s,a)} \left| \mathbb{E}_{a' \sim \pi} \left[ Q_{\mathcal{M}_{\mathrm{tar}}}^{\pi_{D_{\mathrm{tar}}}}(s', a') \right] - \mathbb{E}_{a' \sim \pi_{D_{\mathrm{tar}}}} \left[ Q_{\mathcal{M}_{\mathrm{tar}}}^{\pi_{D_{\mathrm{tar}}}}(s', a') \right] \right|$$

$$\geq -\frac{1}{1-\gamma} \mathbb{E}_{\rho_{\mathcal{M}_{\mathrm{tar}}}^{\pi}(s,a), s' \sim P_{\mathcal{M}_{\mathrm{tar}}}(\cdot|s,a)} \left| \sum_{a' \in \mathcal{A}} (\pi_{D_{\mathrm{tar}}}(a'|s') - \pi(a'|s')) Q_{\mathcal{M}_{\mathrm{tar}}}^{\pi_{D_{\mathrm{tar}}}}(s', a') \right|$$

$$\geq -\frac{r_{\max}}{(1-\gamma)^2} \mathbb{E}_{\rho_{\mathcal{M}_{\mathrm{tar}}}^{\pi}(s,a), s' \sim P_{\mathcal{M}_{\mathrm{tar}}}(\cdot|s,a)} \left| \sum_{a' \in \mathcal{A}} (\pi_{D_{\mathrm{tar}}}(a'|s') - \pi(a'|s')) \right|$$

$$= -\frac{2r_{\max}}{(1-\gamma)^2} \mathbb{E}_{\rho_{\mathcal{M}_{\mathrm{tar}}}^{\pi}(s,a), P_{\mathcal{M}_{\mathrm{tar}}}} [D_{\mathrm{TV}}(\pi_{D_{\mathrm{tar}}}(\cdot|s') \| \pi(\cdot|s'))],$$

where we use the fact that $|Q(s,a)| \leq \frac{r_{\max}}{1-\gamma}$. Then, we can bound term $(c)$ by using Lemma B.2,

$$
\begin{aligned}
(c) &= J_{\widehat{\mathcal{M}}_{\mathrm{src}}}(\pi_{D_{\mathrm{src}}}) - J_{\widehat{\mathcal{M}}_{\mathrm{src}}}(\pi) \\
&= \frac{1}{1-\gamma} \mathbb{E}_{\rho^{\pi_{D_{\mathrm{src}}}}_{\widehat{\mathcal{M}}_{\mathrm{src}}}(s,a), s' \sim P_{\widehat{\mathcal{M}}_{\mathrm{src}}}(\cdot|s,a)} \left[ \mathbb{E}_{a' \sim \pi_{D_{\mathrm{src}}}} \left[ Q^{\pi}_{\widehat{\mathcal{M}}_{\mathrm{src}}}(s',a') \right] - \mathbb{E}_{a' \sim \pi} \left[ Q^{\pi}_{\widehat{\mathcal{M}}_{\mathrm{src}}}(s',a') \right] \right] \\
&\geq -\frac{1}{1-\gamma} \mathbb{E}_{\rho^{\pi_{D_{\mathrm{src}}}}_{\widehat{\mathcal{M}}_{\mathrm{src}}}(s,a), s' \sim P_{\widehat{\mathcal{M}}_{\mathrm{src}}}(\cdot|s,a)} \left| \mathbb{E}_{a' \sim \pi_{D_{\mathrm{src}}}} \left[ Q^{\pi}_{\widehat{\mathcal{M}}_{\mathrm{src}}}(s',a') \right] - \mathbb{E}_{a' \sim \pi} \left[ Q^{\pi}_{\widehat{\mathcal{M}}_{\mathrm{src}}}(s',a') \right] \right| \\
&\geq -\frac{1}{1-\gamma} \mathbb{E}_{\rho^{\pi_{D_{\mathrm{src}}}}_{\widehat{\mathcal{M}}_{\mathrm{src}}}(s,a), s' \sim P_{\widehat{\mathcal{M}}_{\mathrm{src}}}(\cdot|s,a)} \left| \sum_{a' \in \mathcal{A}} (\pi_{D_{\mathrm{src}}}(a'|s') - \pi(a'|s')) Q^{\pi}_{\widehat{\mathcal{M}}_{\mathrm{src}}}(s',a') \right| \\
&\geq -\frac{r_{\max}}{(1-\gamma)^2} \mathbb{E}_{\rho^{\pi_{D_{\mathrm{src}}}}_{\widehat{\mathcal{M}}_{\mathrm{src}}}(s,a), s' \sim P_{\widehat{\mathcal{M}}_{\mathrm{src}}}(\cdot|s,a)} \left| \sum_{a' \in \mathcal{A}} (\pi_{D_{\mathrm{src}}}(a'|s') - \pi(a'|s')) \right| \\
&= -\frac{2r_{\max}}{(1-\gamma)^2} \mathbb{E}_{\rho^{\pi_{D_{\mathrm{src}}}}_{\widehat{\mathcal{M}}_{\mathrm{src}}}(s,a), P_{\widehat{\mathcal{M}}_{\mathrm{src}}}} \left[ D_{\mathrm{TV}}(\pi_{D_{\mathrm{src}}}(\cdot|s') \| \pi(\cdot|s')] \right.
\end{aligned}
$$

Finally, we bound term $(b)$. By using Lemma B.1, we have

$$
\begin{aligned}
(b) &= J_{\mathcal{M}_{\mathrm{tar}}}(\pi_{D_{\mathrm{tar}}}) - J_{\widehat{\mathcal{M}}_{\mathrm{src}}}(\pi_{D_{\mathrm{src}}}) \\
&= -\frac{1}{1-\gamma} \mathbb{E}_{\rho^{\pi_{D_{\mathrm{tar}}}}_{\mathcal{M}_{\mathrm{tar}}}(s,a)} \left[ \mathbb{E}_{s' \sim P_{\mathcal{M}_{\mathrm{tar}}}, a' \sim \pi_{D_{\mathrm{tar}}}} [Q^{\pi_{D_{\mathrm{src}}}}_{\widehat{\mathcal{M}}_{\mathrm{src}}}(s',a')] - \mathbb{E}_{s' \sim P_{\widehat{\mathcal{M}}_{\mathrm{src}}}, a' \sim \pi_{D_{\mathrm{src}}}} [Q^{\pi_{D_{\mathrm{src}}}}_{\widehat{\mathcal{M}}_{\mathrm{src}}}(s',a')] \right] \\
&= -\frac{1}{1-\gamma} \mathbb{E}_{\rho^{\pi_{D_{\mathrm{tar}}}}_{\mathcal{M}_{\mathrm{tar}}}(s,a)} \left[ \underbrace{\left( \mathbb{E}_{s' \sim P_{\mathcal{M}_{\mathrm{tar}}}, a' \sim \pi_{D_{\mathrm{tar}}}} \left[ Q^{\pi_{D_{\mathrm{src}}}}_{\widehat{\mathcal{M}}_{\mathrm{src}}}(s',a') \right] - \mathbb{E}_{s' \sim P_{\mathcal{M}_{\mathrm{tar}}}, a' \sim \pi_{D_{\mathrm{src}}}} \left[ Q^{\pi_{D_{\mathrm{src}}}}_{\widehat{\mathcal{M}}_{\mathrm{src}}}(s',a') \right] \right)}_{(d)} \right. \\
&\qquad \left. + \underbrace{\left( \mathbb{E}_{s' \sim P_{\mathcal{M}_{\mathrm{tar}}}, a' \sim \pi_{D_{\mathrm{src}}}} \left[ Q^{\pi_{D_{\mathrm{src}}}}_{\widehat{\mathcal{M}}_{\mathrm{src}}}(s',a') \right] - \mathbb{E}_{s' \sim P_{\widehat{\mathcal{M}}_{\mathrm{src}}}, a' \sim \pi_{D_{\mathrm{src}}}} \left[ Q^{\pi_{D_{\mathrm{src}}}}_{\widehat{\mathcal{M}}_{\mathrm{src}}}(s',a') \right] \right)}_{(e)} \right].
\end{aligned}
$$

For term $(d)$, it is easy to find that:

$$
\begin{aligned}
(d) &= \mathbb{E}_{s' \sim P_{\mathcal{M}_{\mathrm{tar}}}} \left[ \sum_{a' \in \mathcal{A}} (\pi_{D_{\mathrm{tar}}}(a'|s') - \pi_{D_{\mathrm{src}}}(a'|s')) Q^{\pi_{D_{\mathrm{src}}}}_{\widehat{\mathcal{M}}_{\mathrm{src}}}(s',a') \right] \\
&\leq \mathbb{E}_{s' \sim P_{\mathcal{M}_{\mathrm{tar}}}} \left[ \sum_{a' \in \mathcal{A}} |\pi_{D_{\mathrm{tar}}}(a'|s') - \pi_{D_{\mathrm{src}}}(a'|s')| \times |Q^{\pi_{D_{\mathrm{src}}}}_{\widehat{\mathcal{M}}_{\mathrm{tar}}}(s',a')| \right] \\
&\leq \frac{2r_{\max}}{1-\gamma} \mathbb{E}_{s' \sim P_{\mathcal{M}_{\mathrm{tar}}}} \left[ D_{\mathrm{TV}}(\pi_{D_{\mathrm{tar}}}(\cdot|s') \| \pi_{D_{\mathrm{src}}}(\cdot|s')) \right].
\end{aligned}
$$

It remains to bound term $(e)$. We have

$$
\begin{aligned}
(e) &= \mathbb{E}_{s' \sim P_{\mathcal{M}_{\mathrm{tar}}}, a' \sim \pi_{D_{\mathrm{src}}}} [Q^{\pi_{D_{\mathrm{src}}}}_{\widehat{\mathcal{M}}_{\mathrm{src}}}(s',a')] - \mathbb{E}_{s' \sim P_{\widehat{\mathcal{M}}_{\mathrm{src}}}, a' \sim \pi_{D_{\mathrm{src}}}} [Q^{\pi_{D_{\mathrm{src}}}}_{\widehat{\mathcal{M}}_{\mathrm{src}}}(s',a')] \\
&= \mathbb{E}_{a' \sim \pi_{D_{\mathrm{src}}}} \left[ \sum_{s'} (P_{\mathcal{M}_{\mathrm{tar}}}(s'|s,a) - P_{\widehat{\mathcal{M}}_{\mathrm{src}}}(s'|s,a)) Q^{\pi_{D_{\mathrm{src}}}}_{\widehat{\mathcal{M}}_{\mathrm{src}}}(s',a') \right] \\
&\leq \mathbb{E}_{a' \sim \pi_{D_{\mathrm{src}}}} \left[ \sum_{s'} |P_{\mathcal{M}_{\mathrm{tar}}}(s'|s,a) - P_{\widehat{\mathcal{M}}_{\mathrm{src}}}(s'|s,a)| \times |Q^{\pi}_{\widehat{\mathcal{M}}_{\mathrm{src}}}(s',a')| \right] \\
&\leq \frac{r_{\max}}{1-\gamma} \mathbb{E}_{a' \sim \pi_{D_{\mathrm{src}}}} \left[ \sum_{s'} \left| P_{\mathcal{M}_{\mathrm{tar}}}(s'|s,a) - P_{\widehat{\mathcal{M}}_{\mathrm{src}}}(s'|s,a) \right| \right] \\
&= \frac{2r_{\max}}{1-\gamma} \mathbb{E}_{a' \sim \pi_{D_{\mathrm{src}}}} \left[ D_{\mathrm{TV}}(P_{\mathcal{M}_{\mathrm{tar}}}(\cdot|s,a) \| P_{\widehat{\mathcal{M}}_{\mathrm{src}}}(\cdot|s,a)) \right].
\end{aligned}
$$

Then, we get the bound for term $(b)$:

$$(b) = J_{\mathcal{M}_{\mathrm{tar}}}(\pi_{D_{\mathrm{tar}}}) - J_{\widehat{\mathcal{M}}_{\mathrm{src}}}(\pi_{D_{\mathrm{src}}})$$

$$\geq -\frac{2r_{\max}}{(1-\gamma)^2}\mathbb{E}_{\rho^{\pi_{D_{\mathrm{tar}}}}_{\mathcal{M}_{\mathrm{tar}}}(s,a),s'\sim P_{\mathcal{M}_{\mathrm{tar}}}}\left[D_{\mathrm{TV}}(\pi_{D_{\mathrm{tar}}}(\cdot|s')\|\pi_{D_{\mathrm{src}}}(\cdot|s'))\right]$$

$$-\frac{2r_{\max}}{(1-\gamma)^2}\mathbb{E}_{\rho^{\pi_D}_{\mathcal{M}_{\mathrm{src}}}(s,a),a'\sim\pi_{D_{\mathrm{src}}}}\left[D_{\mathrm{TV}}(P_{\mathcal{M}_{\mathrm{tar}}}(\cdot|s,a)\|P_{\widehat{\mathcal{M}}_{\mathrm{src}}}(\cdot|s,a))\right].$$

Note that the behavior policy in the source domain and the behavior policy in the target domain are fixed, indicating that their total variation deviation under the target domain transition dynamics is constant. Combining the above bounds for term $(a)$, term $(b)$ and term $(c)$, and we have

$$J_{\mathcal{M}_{\mathrm{tar}}}(\pi) - J_{\widehat{\mathcal{M}}_{\mathrm{src}}}(\pi) \geq -\frac{2r_{\max}}{(1-\gamma)^2}\mathbb{E}_{\rho^{\pi}_{\mathcal{M}_{\mathrm{tar}}},P_{\mathcal{M}_{\mathrm{tar}}}}\left[D_{\mathrm{TV}}(\pi_{D_{\mathrm{tar}}}\|\pi)\right] - \frac{2r_{\max}}{(1-\gamma)^2}\mathbb{E}_{\rho^{\pi_{D_{\mathrm{src}}}}_{\widehat{\mathcal{M}}_{\mathrm{src}}},P_{\widehat{\mathcal{M}}_{\mathrm{src}}}}\left[D_{\mathrm{TV}}(\pi_{D_{\mathrm{src}}}\|\pi)\right]$$

$$-\frac{2r_{\max}}{(1-\gamma)^2}\mathbb{E}_{\rho^{\pi_D}_{\mathcal{M}_{\mathrm{src}}},\pi_{D_{\mathrm{src}}}}\left[D_{\mathrm{TV}}(P_{\mathcal{M}_{\mathrm{tar}}}\|P_{\widehat{\mathcal{M}}_{\mathrm{src}}})\right] - \text{constant}.$$

By replacing $\dfrac{2r_{\max}}{(1-\gamma)^2}$ with $C_1$, we have the desired conclusion immediately. $\qquad\square$

### B.3 PROOF OF THEOREM 3.2

**Theorem B.4.** *Assume that the cost is bounded, i.e., $C(u, u') \leq C_{\max} < \infty, \forall\, u, u'$, then we have*

$$0 \geq d(u_t) \geq -C_{\max}D_{\mathrm{TV}}(p_s\|p_t).$$

*Proof.* Recall that $p_s = \frac{1}{|D_{\mathrm{src}}|}\sum_{t=1}^{|D_{\mathrm{src}}|}\delta_{u_t}$ and $p_t = \frac{1}{|D_{\mathrm{tar}}|}\sum_{t=1}^{|D_{\mathrm{tar}}|}\delta_{u'_t}$, where $u = s_{\mathrm{src}}\oplus a_{\mathrm{src}}\oplus s'_{\mathrm{src}}$, $u' = s_{\mathrm{tar}}\oplus a_{\mathrm{tar}}\oplus s'_{\mathrm{tar}}$, and $(s_{\mathrm{src}}, a_{\mathrm{src}}, s'_{\mathrm{src}}) \sim D_{\mathrm{src}}, (s_{\mathrm{tar}}, a_{\mathrm{tar}}, s'_{\mathrm{tar}}) \sim D_{\mathrm{tar}}$.

Note that the cost $C(u, u') \geq 0, \forall\, u, u'$ and the solved optimal coupling $\mu^*$ is also non-negative. It is then easy to find that

$$d(u_t) = -\sum_{t'=1}^{|D_{\mathrm{tar}}|} C(u_t, u'_{t'})\mu^*_{t,t'} \leq 0, \quad u_t = (s^t_{\mathrm{src}}, a^t_{\mathrm{src}}, (s'_{\mathrm{src}})^t) \sim D_{\mathrm{src}}. \tag{15}$$

We also have that

$$-\sum_{t=1}^{|D_{\mathrm{src}}|} d_t = \mathcal{W}(u, u') = \sum_{t=1}^{|D_{\mathrm{src}}|}\sum_{t'=1}^{|D_{\mathrm{tar}}|} C(u_t, u'_{t'})\mu^*_{t,t'}. \tag{16}$$

Using the definition of the Wasserstein distance, we have

$$-\sum_{t=1}^{|D_{\mathrm{src}}|} d_t = \min_{\mu\in M}\sum_{t=1}^{|D_{\mathrm{src}}|}\sum_{t'=1}^{|D_{\mathrm{tar}}|} C(u_t, u'_{t'})\mu_{t,t'}$$

$$\leq \min_{\mu\in M}\max_{t,t'} C(u_t, u'_{t'})\sum_{t=1}^{|D_{\mathrm{src}}|}\sum_{t'=1}^{|D_{\mathrm{tar}}|} \mathbb{1}(u_t \neq u_{t'})\mu_{t,t'}$$

$$= C_{\max}\min_{\mu\in M}\sum_{t=1}^{|D_{\mathrm{src}}|}\sum_{t'=1}^{|D_{\mathrm{tar}}|} \mathbb{1}(u_t \neq u_{t'})\mu_{t,t'}$$

$$= C_{\max}D_{\mathrm{TV}}(p_s\|p_t),$$

where the last equation holds due to Remark 2.27 in Peyré & Cuturi (2019). Finally, we have

$$-d_t \leq -\sum_{t=1}^{|D_{\mathrm{src}}|} d_t \leq C_{\max}D_{\mathrm{TV}}(p_s\|p_t) \Leftrightarrow d_t \geq -C_{\max}D_{\mathrm{TV}}(p_s\|p_t). \tag{17}$$

This concludes the proof. $\qquad\square$

## C  ENVIRONMENT SETTING

In this section, we include a detailed description of the environmental settings we adopt in this paper, such as the basic information on the source domain datasets and target domain datasets, details on how dynamics shifts are realized, etc.

### C.1  DATASETS

**Source domain datasets.** Since we consider the setting where only a few target domain data are available, we directly adopt the MuJoCo datasets from D4RL (Fu et al., 2020) as *source domain datasets*. In D4RL, the MuJoCo datasets are collected during the interactions with the continuous action environments in Gym (Brockman et al., 2016) simulated by MuJoCo (Todorov et al., 2012). We adopt four tasks, *halfcheetah, hopper, walker2d, ant*, and consider source domain dataset qualities across *medium, medium-replay, medium-expert*. The **medium** datasets contain experiences collected from an early-stopped SAC policy for 1M steps. The **medium-replay** datasets record the replay buffer of a policy trained up to the performance of the medium agent. The **medium-expert** datasets are constructed by mixing the medium data and expert data at a 50-50 ratio. Note that the source domain datasets can have quite distinct dataset sizes, *e.g.*, **medium** datasets have 1M samples while **medium-replay** datasets can only have 100K samples.

**Target domain datasets.** To examine the offline policy adaptation capability of our method, we design three kinds of dynamics shift scenarios for empirical evaluations based on four widely used MuJoCo tasks (*HalfCheetah-v3*, *Hopper-v3*, *Walker2d-v3*, *Ant-v3*), including *gravity shift*, *kinematic shift* and *morphology shift*. The gravity shift means that the gravitational forces acting on the robot in the source and target domains are different, the kinematic shift indicates that some joints of the simulated robot are broken in the target domain, while the morphology shift suggests that there are some morphological mismatch between the simulated robot in two domains. We show the visualization results of the agent in the source domain and the target domain in Figure 4. We explicate the detailed modifications in the following subsections.

We then collect target domain datasets in the revised environments by following a similar data-collecting procedure as D4RL. We train an SAC (Haarnoja et al., 2018) agents in environments with dynamics shifts for 1M steps. We log the policy checkpoints of the agent during training and use them for rolling out trajectories. The **expert** datasets are generated using the last policy checkpoint, and the **medium** datasets are gathered with the policy checkpoint that exhibits approximately 1/2 or 1/3 the performance of the expert policy. To ensure that only a limited budget of target domain data can be accessed, we only collect 5 trajectories for each dataset, *which amounts to about 5000 transitions*. We do not follow D4RL to simply merge the medium dataset and the expert dataset to produce **medium-expert** datasets. Instead, we strictly follow the data budget and pick 2 trajectories from the medium dataset and 3 trajectories from the expert dataset to construct the **medium-expert** datasets. We observe that it is extremely difficult for off-the-shelf offline RL methods to acquire meaningful performance given such little data. Since we evaluate the performance of the agent in our modified environments, we follow D4RL and propose to use the *normalized score* metric to better characterize the performance of the agent across different tasks. The normalized score performance of the agent in the target domain gives:

$$NS = \frac{J - J_r}{J_e - J_r} \times 100, \qquad (18)$$

where $J$ is the return acquired by the agent in the target domain, $J_r, J_e$ are the returns obtained by the random policy and the expert policy in the target domain, respectively. We summarize the reference scores of $J_r$ and $J_e$ under different dynamics shift scenarios in Table 3. We also list the minimum return, the maximum return, and the average return of trajectories in each target domain offline dataset in Table 4.

### C.2  GRAVITY SHIFT TASKS

To simulate the gravity shifts between the source domain and the target domain, we modify the `xml` files of the underlying environments. We set the gravitational acceleration of the target domain to half of that in the source domain, and we do not alter the direction of the gravitational force.

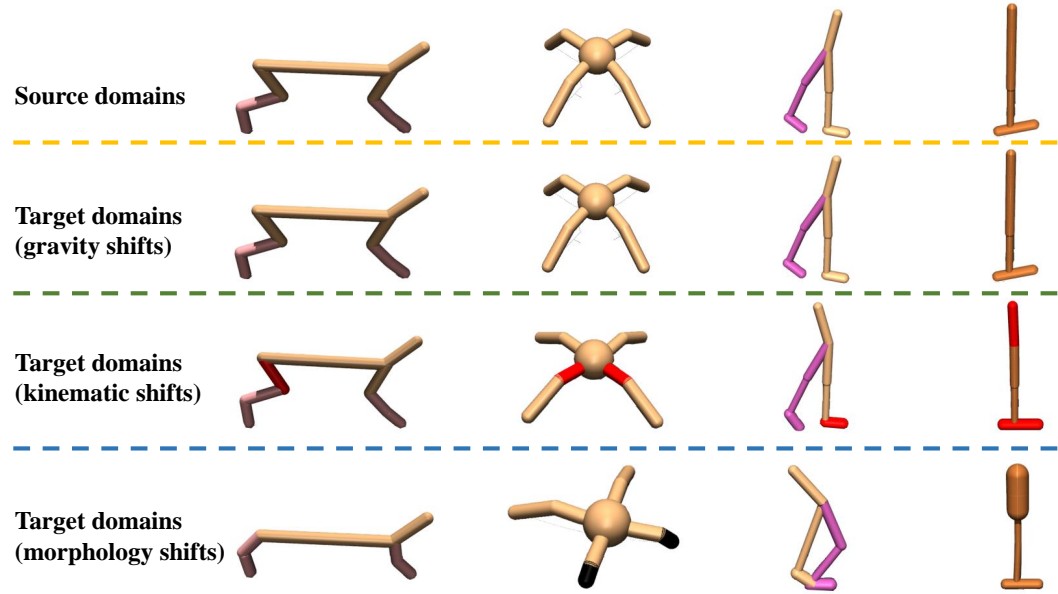

Figure 4: **Illustration of the adopted environments.** Target domain robots can have gravity shifts (*second row*), kinematic shifts (*third row*), and morphology shifts (*bottom*) compared with the source domain (*top*) robots.

Table 3: **The referenced min score and max score for MuJoCo datasets under various dynamics shift scenarios.** These are used to evaluate the normalized score performance of the algorithm in the target domain. Since the source domain and the target domain differ from each other in transition dynamics, one could not simply adopt the reference scores from D4RL directly.

| Task Name | Dynamics shift type | Reference min score $J_r$ | Reference max score $J_e$ |
|---|---|---|---|
| halfcheetah | gravity | $-280.18$ | 9509.15 |
| halfcheetah | kinematic | $-280.18$ | 7065.03 |
| halfcheetah | morphology | $-280.18$ | 9713.59 |
| hopper | gravity | $-26.336$ | 3234.3 |
| hopper | kinematic | $-26.336$ | 2842.73 |
| hopper | morphology | $-26.336$ | 3152.75 |
| walker2d | gravity | 10.08 | 5194.713 |
| walker2d | kinematic | 10.08 | 3257.51 |
| walker2d | morphology | 10.08 | 4398.43 |
| ant | gravity | $-325.6$ | 4317.065 |
| ant | kinematic | $-325.6$ | 5122.57 |
| ant | morphology | $-325.6$ | 5722.01 |

*halfcheetah / hopper / walker2d / ant-gravity*: the modifications of the `xml` file gives:

```
# gravity
<option gravity="0 0 -4.905" timestep="0.01"/>
```

### C.3 KINEMATIC SHIFT TASKS

Different from the gravity shift, the kinematic shift occurs at different parts of the simulated robot. Detailed modifications of the `xml` files give:

*halfcheetah-kinematic*: The rotation angle of the joint on the thigh of the robot's back leg is modified from $[-0.52, 1.05]$ to $[-0.0052, 0.0105]$.

Table 4: **Trajectory return information of the target domain datasets.** We list here the minimum return (abbreviated as *min return*), the maximum return (denoted as *max return*), and the average return of trajectories in the target domain datasets.

| Task Name | Dynamics shift | Dataset type | Min return | Max return | Average return |
|---|---|---|---|---|---|
| halfcheetah | gravity | medium | 4179.82 | 4383.32 | 4296.27 |
| halfcheetah | gravity | medium-expert | 4342.78 | 8243.03 | 6567.94 |
| halfcheetah | gravity | expert | 7846.18 | 8339.18 | 8131.54 |
| halfcheetah | kinematic | medium | 2709.52 | 2782.61 | 2755.50 |
| halfcheetah | kinematic | medium-expert | 2709.52 | 7065.04 | 5298.61 |
| halfcheetah | kinematic | expert | 6951.27 | 7065.04 | 6998.93 |
| halfcheetah | morphology | medium | 4070.43 | 4214.64 | 4156.88 |
| halfcheetah | morphology | medium-expert | 4070.43 | 9713.60 | 7475.13 |
| halfcheetah | morphology | expert | 9474.34 | 9719.81 | 9614.92 |
| hopper | gravity | medium | 1784.88 | 2885.13 | 2367.66 |
| hopper | gravity | medium-expert | 2416.82 | 4143.63 | 3297.79 |
| hopper | gravity | expert | 3745.59 | 4186.19 | 4051.07 |
| hopper | kinematic | medium | 1849.06 | 1886.89 | 1870.16 |
| hopper | kinematic | medium-expert | 1868.20 | 2842.17 | 2452.67 |
| hopper | kinematic | expert | 2840.97 | 2842.73 | 2841.83 |
| hopper | morphology | medium | 1946.73 | 2039.80 | 1980.33 |
| hopper | morphology | medium-expert | 1980.06 | 3152.75 | 2694.58 |
| hopper | morphology | expert | 3148.26 | 3152.75 | 3151.39 |
| walker2d | gravity | medium | 2421.98 | 3444.63 | 2897.85 |
| walker2d | gravity | medium-expert | 3144.32 | 5166.62 | 4415.11 |
| walker2d | gravity | expert | 5159.51 | 5219.14 | 5174.51 |
| walker2d | kinematic | medium | 1415.69 | 2223.17 | 2026.49 |
| walker2d | kinematic | medium-expert | 1415.69 | 3257.51 | 2442.82 |
| walker2d | kinematic | expert | 2874.92 | 3257.51 | 3077.19 |
| walker2d | morphology | medium | 772.82 | 2875.40 | 2013.11 |
| walker2d | morphology | medium-expert | 772.82 | 4348.94 | 2961.67 |
| walker2d | morphology | expert | 4341.38 | 4398.44 | 4354.58 |
| ant | gravity | medium | 377.10 | 3247.66 | 2314.45 |
| ant | gravity | medium-expert | 377.10 | 4511.55 | 2131.79 |
| ant | gravity | expert | 335.28 | 4584.53 | 3365.35 |
| ant | kinematic | medium | 2826.00 | 3111.93 | 3017.82 |
| ant | kinematic | medium-expert | 2826.00 | 5122.58 | 4240.99 |
| ant | kinematic | expert | 5009.82 | 5122.57 | 5072.50 |
| ant | morphology | medium | 2150.50 | 2204.32 | 2176.02 |
| ant | morphology | medium-expert | 5150.50 | 5653.05 | 4199.54 |
| ant | morphology | expert | 5461.49 | 5722.01 | 5586.73 |

```
# broken back thigh
<joint axis="0 1 0" damping="6" name="bthigh" pos="0 0 0" range="-.0052
    .0105" stiffness="240" type="hinge"/
```

***hopper-kinematic***: The rotation angle of the head joint is modified from $[-150, 0]$ to $[-0.15, 0]$ and the rotation angle of the foot joint is modified from $[-45, 45]$ to $[-18, 18]$.

```
# head joint
<joint axis="0 -1 0" name="thigh_joint" pos="0 0 1.05" range="-0.15 0"
    type="hinge"/>
# foot joint
<joint axis="0 -1 0" name="foot_joint" pos="0 0 0.1" range="-18 18" type=
    "hinge"/>
```

*walker-kinematic*: We modify the rotation angle of the foot joint on the robot's right leg from $[-45, 45]$ to $[-0.45, 0.45]$.

```
# right foot
<joint axis="0 -1 0" name="foot_joint" pos="0 0 0.1" range="-0.45 0.45"
    type="hinge"/>
```

*ant-kinematic*: The rotation angles of the joints on the hip of two legs in the ant robot are modified from $[-30, 30]$ to $[-0.3, 0.3]$

```
# hip joints of front legs
<joint axis="0 0 1" name="hip_1" pos="0.0 0.0 0.0" range="-0.3 0.3" type=
    "hinge"/>
<joint axis="0 0 1" name="hip_2" pos="0.0 0.0 0.0" range="-0.3 0.3" type=
    "hinge"/>
```

## C.4 MORPHOLOGY SHIFT TASKS

Akin to the kinematic shift tasks, the morphological change of the robot in the target domain differs per environment. To be specific,

*halfcheetah-morph*: We modify the sizes of the back thigh and the forward thigh of the Cheetah robot as below:

```
# back thigh
<geom fromto="0 0 0 -0.0001 0 -0.0001" name="bthigh" size="0.046" type="
    capsule"/>
<body name="bshin" pos="-0.0001 0 -0.0001">
# forward thigh
<geom fromto="0 0 0 0.0001 0 0.0001" name="fthigh" size="0.046" type="
    capsule"/>
<body name="fshin" pos="0.0001 0 0.0001">
```

*hopper-morph*: We increase the head size of the robot. The modifications are shown below:

```
# head size
<geom friction="0.9" fromto="0 0 1.45 0 0 1.05" name="torso_geom" size="
    0.125" type="capsule"/>
```

*walker-morph*: We modify the thigh on the right leg of the robot as the following:

```
# right leg
<body name="thigh" pos="0 0 1.05">
 <joint axis="0 -1 0" name="thigh_joint" pos="0 0 1.05" range="-150 0"
     type="hinge"/>
 <geom friction="0.9" fromto="0 0 1.05 0 0 1.045" name="thigh_geom" size=
     "0.05" type="capsule"/>
 <body name="leg" pos="0 0 0.35">
   <joint axis="0 -1 0" name="leg_joint" pos="0 0 1.045" range="-150 0"
       type="hinge"/>
   <geom friction="0.9" fromto="0 0 1.045 0 0 0.3" name="leg_geom" size="
       0.04" type="capsule"/>
   <body name="foot" pos="0.2 0 0">
     <joint axis="0 -1 0" name="foot_joint" pos="0 0 0.3" range="-45 45"
         type="hinge"/>
     <geom friction="0.9" fromto="-0.0 0 0.3 0.2 0 0.3" name="foot_geom"
         size="0.06" type="capsule"/>
   </body>
 </body>
</body>
```

*ant-morph*: We reduce the size of the ant robot's feet on its front two legs to simulate an ant robot with short feet. The detailed modifications are:

```
# front leg 1
<geom fromto="0.0 0.0 0.0 0.1 0.1 0.0" name="left_ankle_geom" size="0.08"
     type="capsule"/>
# front leg 2
<geom fromto="0.0 0.0 0.0 -0.1 0.1 0.0" name="right_ankle_geom" size="
    0.08" type="capsule"/>
```

We have included the modified `xml` files in the supplementary material.

## D  IMPLEMENTATION DETAILS

In this part, we describe the implementation details of the baseline methods and our proposed OTDF algorithm. We also provide a detailed pseudocode for OTDF+IQL. Furthermore, we list the hyper-parameter setup used for all methods.

### D.1  BASELINES

**IQL\*:** IQL (Kostrikov et al., 2022) is a widely used offline RL algorithm that learns an *in-sample* policy without querying OOD samples that lie outside of the offline datasets. Since we observe that IQL can not learn meaningful policies merely on the target domain dataset, we slightly alter its training objectives by involving both the source domain dataset and the target domain dataset. We name the revised method IQL\*. It trains the state value function via expectile regression:

$$\mathcal{L}_V = \mathbb{E}_{(s,a)\sim D_{\mathrm{src}}\cup D_{\mathrm{tar}}}[L_2^\tau(Q_{\theta'}(s,a) - V_\psi(s))], \tag{19}$$

where $L_2^\tau(u) = |\tau - \mathbb{1}(u < 0)|u^2$, $\mathbb{1}(\cdot)$ is the indicator function, $\theta'$ is the target network parameter. The state-action value function is then updated by:

$$\mathcal{L}_Q = \mathbb{E}_{(s,a,r,s')\sim D_{\mathrm{src}}\cup D_{\mathrm{tar}}}[(r(s,a) + \gamma V_\psi(s') - Q_\theta(s,a))^2]. \tag{20}$$

We then calculate the advantage function $A(s,a) = Q(s,a) - V(s)$ and use it as weights for policy optimization:

$$\mathcal{L}_{\mathrm{actor}} = \mathbb{E}_{(s,a)\sim D_{\mathrm{src}}\cup D_{\mathrm{tar}}}\left[\exp(\beta_{\mathrm{IQL}} \times A(s,a))\log \pi_\phi(a|s)\right], \tag{21}$$

where $\beta_{\mathrm{IQL}}$ is the inverse temperature coefficient. We implement IQL by following its official codebase[2]. We adopt the symmetric sampling method when sampling data from two offline datasets.

**DARA:** DARA (Liu et al., 2022a) is the offline version of DARC (Eysenbach et al., 2021). It trains two domain classifiers $q_{\theta_{\mathrm{SAS}}}(\mathrm{target}|s_t, a_t, s_{t+1})$, $q_{\theta_{\mathrm{SA}}}(\mathrm{target}|s_t, a_t)$ with the following objectives

$$\mathcal{L}(\theta_{\mathrm{SAS}}) = \mathbb{E}_{D_{\mathrm{tar}}}\left[\log q_{\theta_{\mathrm{SAS}}}(\mathrm{target}|s_t, a_t, s_{t+1})\right] + \mathbb{E}_{D_{\mathrm{src}}}\left[\log(1 - q_{\theta_{\mathrm{SAS}}}(\mathrm{target}|s_t, a_t, s_{t+1}))\right],$$
$$\mathcal{L}(\theta_{\mathrm{SA}}) = \mathbb{E}_{D_{\mathrm{tar}}}\left[\log q_{\theta_{\mathrm{SA}}}(\mathrm{target}|s_t, a_t)\right] + \mathbb{E}_{D_{\mathrm{src}}}\left[\log(1 - q_{\theta_{\mathrm{SA}}}(\mathrm{target}|s_t, a_t))\right],$$

to estimate the dynamics gap $\log \frac{P_{\mathcal{M}_{\mathrm{tar}}}(s_{t+1}|s_t,a_t)}{P_{\mathcal{M}_{\mathrm{src}}}(s_{t+1}|s_t,a_t)}$ between the source domain and the target domain. DARA approximates this term by leveraging the trained classifiers and proposes to modify the source domain rewards as follows

$$\hat{r}_{\mathrm{DARA}} = r - \lambda \times \delta_r, \quad \delta r(s_t, a_t) = -\log \frac{q_{\theta_{\mathrm{SAS}}}(\mathrm{target}|s_t, a_t, s_{t+1})q_{\theta_{\mathrm{SA}}}(\mathrm{source}|s_t, a_t)}{q_{\theta_{\mathrm{SAS}}}(\mathrm{source}|s_t, a_t, s_{t+1})q_{\theta_{\mathrm{SA}}}(\mathrm{target}|s_t, a_t)}, \tag{22}$$

where $\lambda$ is an important hyperparameter that controls the strengths of the reward penalty. We empirically find that setting $\lambda = 1$ or a larger value leads to often incurs quite poor performance. We hence set $\lambda = 0.1$ for DARA by default. We implement DARA by following its code attached in https://openreview.net/forum?id=9SDQB3b68K. We clip the reward penalty term to lie in $[-10, 10]$ for training stability. We use IQL as the base algorithm for DARA to align with other algorithms.

**BOSA:** BOSA (Liu et al., 2024a) proposes to address the cross-domain offline RL problems by including two support constraints, which handle the OOD state actions problem through a supported

---

[2]https://github.com/ikostrikov/implicit_q_learning

policy optimization and mitigate the OOD dynamics issue through a supported value optimization. To be specific, the critics in BOSA are updated via:

$$\mathcal{L}_{\text{critic}} = \underset{(s,a)\sim D_{\text{src}}}{\mathbb{E}}[Q_{\theta_i}(s,a)] + \underset{\substack{(s,a,r,s')\sim D_{\text{src}}\cup D_{\text{tar}}, \\ a'\sim \pi_\phi(\cdot|s)}}{\mathbb{E}}\left[\mathbb{1}(\hat{P}_{\text{tar}}(s'|s,a)) > \epsilon)(Q_{\theta_i}(s,a) - y)^2\right],$$

(23)

where $\mathbb{1}(\cdot)$ is the indicator function, $\hat{P}_{\text{tar}}(s'|s,a) = \arg\max \mathbb{E}_{(s,a,s')\sim D_{\text{tar}}}[\log \hat{P}_{\text{tar}}(s'|s,a)]$ is the estimated target domain transition dynamics, $\epsilon$ is the selection threshold, $i \in \{1,2\}$. The policy in BOSA is updated via another supported optimization objective:

$$\mathcal{L}_{\text{actor}} = \mathbb{E}_{s\sim D_{\text{src}}\cup D_{\text{tar}}, a\sim \pi_\phi(s)}[Q_{\theta_i}(s,a)], \text{ s.t. } \mathbb{E}_{s\sim D_{\text{src}}\cup D_{\text{tar}}}[\hat{\pi}_{\phi_{\text{mix}}}(\pi_\phi(s)|s)] > \epsilon' \quad (24)$$

where $\epsilon'$ is a hyperparameter that determines the selection threshold, $\hat{\pi}_{\phi_{\text{mix}}}$ is the learned behavior policy of the mixed dataset $D_{\text{src}} \cup D_{\text{tar}}$. BOSA models the transition dynamics model in the target domain and the behavior policy of the mixed dataset with CVAE. Since no open-source codes can be found for BOSA, we implement it by following the instructions in the original paper. BOSA is trained for 1M gradient steps in practice by drawing samples from both the source domain dataset and the target domain dataset. We use SPOT (Wu et al., 2022a) as the backbone for BOSA.

**SRPO:** SRPO (Xue et al., 2024) proposes to optimize the policy by solving the following constrained optimization problem:

$$\max_\pi \mathbb{E}_{s_t,a_t\sim\tau_\pi}\left[\sum_{t=0}^\infty \gamma^t r(s_t,a_t)\right] \text{ s.t. } D_{\text{KL}}(d_\pi(\cdot)\|\zeta(\cdot)) < \epsilon, \quad (25)$$

where $\tau_\pi$ is the trajectory induced by the policy $\pi$, $d_\pi(\cdot)$ is the stationary state distribution of policy $\pi$, $\zeta(\cdot)$ denotes the optimal state distribution in other environment dynamics. The above problem can be transformed into the unconstrained optimization problem via Lagrange multipliers, where the logarithm of probability density ratio $\lambda \log \frac{\zeta(s_t)}{d_\pi(s_t)}$ is added to the vanilla reward term. In light of this, SRPO samples a batch of data with size $N$ from two offline datasets $D_{\text{src}}, D_{\text{tar}}$ and ranks these transitions by state values. SRPO then tags a proportion of $\rho N$ samples with high state-values as *real* data and others as *fake* data. It trains a discriminator $D_\delta(\cdot)$ to distinguish these samples, and proposes to modify the rewards via:

$$\hat{r}_{\text{SRPO}} = r + \lambda \times \frac{D_\delta(s)}{1 - D_\delta(s)}, \quad (26)$$

where $\lambda$ is the reward coefficient. We use a fixed proportion of real data $\rho = 0.5$ for all experiments. We do not the find official code for SRPO and implement it ourselves by following its original paper.

**IGDF:** IGDF (Wen et al., 2024) captures the dynamics gap between the source domain and the target domain through contrastive learning. It trains a score function $h(\cdot)$ using $(s,a,s'_{\text{tar}}) \sim D_{\text{tar}}$ from the target domain as positive samples, and mixed transition $(s,a,s'_{\text{src}})$ as negative samples, where $(s,a) \sim D_{\text{tar}}, s'_{\text{src}} \sim D_{\text{src}}$. The score function is optimized via the following contrastive learning objective:

$$\mathcal{L}_{\text{contrastive}} = -\mathbb{E}_{(s,a,s'_{\text{tar}})}\mathbb{E}_{S'^-}\left[\log \frac{h(s,a,s'_{\text{tar}})}{\sum_{s'\sim S'^- \cup s'_{\text{tar}}} h(s,a,s')}\right], \quad (27)$$

where $S'^-$ is a collection of the next states in negative samples. Practically, IGDF adopts two neural networks $\phi(s,a), \psi(s')$ to learn representations of state-action pairs and states, and approximate the score function with a linear parameterization of them:

$$h(s,a,s') = \exp(\phi(s,a)^T \psi(s')). \quad (28)$$

Based on the measured score function, IGDF proposes to filter out source domain data when training value functions:

$$\mathcal{L}_{\text{critic}} = \frac{1}{2}\mathbb{E}_{D_{\text{tar}}}\left[(Q_\theta - \mathcal{T}Q_\theta)^2\right] + \frac{1}{2}\alpha\cdot h(s,a,s')\mathbb{E}_{(s,a,s')\sim D_{\text{src}}}\left[\mathbb{1}(h(s,a,s') > h_{\xi\%})(Q_\theta - \mathcal{T}Q_\theta)^2\right],$$

(29)

where $\alpha$ is the importance coefficient for weighting the TD error of the source domain data, $\xi$ is the data selection ratio akin to that in OTDF. We run IGDF by using its official codebase[3] and use IQL as its backbone.

---

[3] https://github.com/BattleWen/IGDF

### D.2 Algorithmic Details of OTDF

To avoid solving repetitive OT problems across different seeds of the same task, we solve the OT problem before the policy training process begins. This is feasible thanks to the fact that both the source domain and the target domain are offline. We use the cosine distance as the cost function in OTT-JAX. After calculating the optimal coupling, we measure the deviation of the source domain data to the entire target domain dataset via Equation 5. This process is very computationally efficient thanks to the OTT-JAX library, solving the entropy-regularized OT problem within 5 minutes given 1M source domain data and 5000 target domain data. Then we incorporate these deviations into the source domain dataset. For practical usage, we build the practical OTDF algorithm on top of IQL.

We summarize the pseudocode of OTDF+IQL in Algorithm 2. When updating the value function, we selectively reject some source domain data that deviate far from the span of the target domain dataset. We further normalize the deviation obtained by OT with min-max normalization and leverage them as weights for source domain data. This encourages the agent to *adaptively* emphasize source domain data that lie close to the target domain while de-emphasizing others. This can also be viewed as a *regularization* term that penalizes the source domain transitions. Note that the data filtering process only occurs at the optimization process of the *state-action value function $Q$* and the update formula of the *state-value function $V$* remains unchanged.

---

**Algorithm 2** Optimal Transport Data Filtering (OTDF)

---

**Input:** Source domain dataset $D_{\text{src}}$, target domain dataset $D_{\text{tar}}$, batch size $N$, data selection ratio $\xi$

1: Initialize policy network $\pi_\phi$, value networks $V_\psi, Q_\theta$, target $Q$ function $Q_{\theta'}$, the cost function $C$, CVAE $G$, policy coefficients $\beta$, number of sampled latent variables $M$, target update rate $\eta$
2:   // Solve the OT problem
3: Compute the optimal alignment between $D_{\text{src}}$ and $D_{\text{tar}}$ with Equation 4
4: Compute deviations $\{d_t\}_{t=1}^{|D_{\text{src}}|}$ between the source domain data and $D_{\text{tar}}$ with Equation 5
5: Concatenate $D_{\text{src}}$ and $\{d_t\}_{t=1}^{|D_{\text{src}}|}$ to get $\widehat{D}_{\text{src}} = \{(s_t, a_t, r_t, s_t', d_t))\}_{t=1}^{|D_{\text{src}}|}$
6:   // CVAE training
7: Train CVAE policy to model the behavior policy in the target domain dataset with Equation 10
8: **for** $i = 1, 2, ...$ **do**
9:     Sample a mini-batch $b_{\text{src}} := \{(s, a, r, s', d)\}$ with size $\frac{N}{2}$ from $D_{\text{src}}$
10:    Sample a mini-batch $b_{\text{tar}} := \{(s, a, r, s')\}$ with size $\frac{N}{2}$ from $D_{\text{tar}}$
11:    Update the state value function $V_\psi$ via: $\mathcal{L}_V = \mathbb{E}_{(s,a) \sim D_{\text{src}} \cup D_{\text{tar}}}[L_2^\tau(Q_{\theta'}(s,a) - V_\psi(s))]$
12:    Normalize the deviations $d$ via Equation 8 to obtain normalized deviations $\widehat{d}$
13:    // Data filtering
14:    Rank the deviations of the sampled source domain data and admit top $\xi\%$ of them
15:    Compute the weights for the remaining source domain data via $\exp(\widehat{d})$
16:    Compute the target value via: $y = r + \gamma V_\psi(s')$
17:    Optimize the state-action value function $Q_\theta$ on $b_{\text{src}} \cup b_{\text{tar}}$ via:

$$\mathcal{L}_Q = \mathbb{E}_{D_{\text{tar}}}\left[(Q_\theta - y)^2\right] + \mathbb{E}_{(s,a,r,s',d) \sim \widehat{D}_{\text{src}}}\left[\exp(\widehat{d})\mathbb{1}(d > d_{\xi\%})(Q_\theta - y)^2\right].$$

18:    Update the target network via: $\theta' \leftarrow \eta\theta + (1 - \eta)\theta'$
19:    // Dataset regularization
20:    Decode $M$ actions from CVAE and construct Gaussian distributions $\{\widehat{\pi}_{\text{tar}}^i(\cdot|s)\}_{i=1}^M$
21:    Compute the advantage $A$ and optimize the policy $\pi_\phi$ on $b_{\text{src}} \cup b_{\text{tar}}$ using advantage-weighted regression (AWR) and dataset regularization:

$$\mathcal{L}_\pi = \mathbb{E}_{(s,a) \sim D_{\text{src}} \cup D_{\text{tar}}}\left[\exp(\beta_{\text{IQL}} \times A)\log\pi_\phi(a|s)\right] - \beta \times \mathbb{E}_{s \sim D_{\text{src}} \cup D_{\text{tar}}}\log\left[\sum_{i=1}^M \widehat{\pi}_{\text{tar}}^i(\pi(\cdot|s)|s)\right],$$

22: **end for**

---

As for the policy update, we introduce a novel dataset regularization term in conjunction with the vanilla weighted behavior cloning term in IQL. We adopt the conditional variational autoencoder (CVAE) to approximate the behavior policy in the target domain dataset. Note that we find that

CVAE can well-model the target domain behavior policy even given limited target domain data. This phenomenon can also be observed in the single-domain model-free RL, *e.g.*, BCQ (Fujimoto et al., 2019) can achieve quite good performance on Adroit *human* datasets from D4RL. The CVAE $G$ contains an encoder $E$ and a decoder $D$. Both the encoder and the decoder contain two intermediate layers with 750 hidden units in each layer. We use the *relu* activation for each intermediate layer.

We note that the dataset regularization term in Equation 11 can degenerate into:

$$\widehat{\mathcal{L}}_\pi = \mathcal{L}_\pi - \beta \times \mathop{\mathbb{E}}_{s \sim D_{\text{src}} \cup D_{\text{tar}}} \log \left[ \sum_{i=1}^{M} \exp(\log \widehat{\pi}_{\text{tar}}^i(\pi(\cdot|s)|s)) \right]$$

$$= \mathcal{L}_\pi - \beta \times \mathop{\mathbb{E}}_{s \sim D_{\text{src}} \cup D_{\text{tar}}} \log \left[ \sum_{i=1}^{M} \widehat{\pi}_{\text{tar}}^i(\pi(\cdot|s)|s) \right].$$

### D.3  Hyperparameter Setup

We summarize the detailed hyperparameter setup for all baseline methods and OTDF in Table 5. For SRPO, we report its best performance by sweeping its reward coefficient $\lambda$ across $\{0.1, 0.3\}$. For IGDF, we set its data selection ratio $\xi\% = 75\%$ as we find setting it to be $25\%$ incurs poor performance. We sweep the representation dimension in IGDF across $\{16, 64\}$ and report the best score. As for OTDF, we adopt a fixed $\xi\% = 80\%$ for all tasks. We set the policy coefficient $\beta = 0.5$ for all of our experiments except for all *halfcheetah* and *walker2d* tasks under *gravity* shifts, where we use $\beta = 0.1$. We do not bother tuning the hyperparameters and demonstrate that our method can achieve quite strong performance with one set of hyperparameters under many scenarios.

## E  Wider Experimental Results

In this section, we provide wider experimental results that are omitted from the main text due to the page limit. We present the comprehensive normalized score comparison of OTDF against other baselines under tasks with kinematic shifts. We also investigate whether the advantages of OTDF can still hold when expert-level source domain datasets are provided. Furthermore, we study whether it is necessary to adaptively weight source domain data.

### E.1  Missing results under kinematic shifts

We summarize the normalized score comparison of OTDF against other baselines under the kinematic shift tasks in Table 6. As shown, OTDF achieves the best performance on **22** out of 36 tasks while remaining competitive against other methods on the rest of the tasks. OTDF achieves a total normalized score of **1547.6**, surpassing IQL* by **29.7%** and the second best approach (IGDF) by **21.8%**. Again, we observe that existing cross-domain offline RL methods often fail to bring performance improvement compared to IQL*, and OTDF is the only algorithm that significantly outperforms them. We believe these further verify the superior offline policy adaptation capability of our proposed OTDF algorithm.

### E.2  Can OTDF beat baselines given high-quality source domain datasets?

In the main text, we only consider source domain datasets with *medium*, *medium-replay* and *medium-expert* qualities. These datasets may typically have broader coverage and possibly contain many transitions that are similar to those in the target domain datasets. However, it can also happen in real-world applications that we may get access to sufficient expert source domain datasets. The expert source domain datasets often have a narrow distribution and state-action coverage. It is then interesting to examine whether OTDF can still beat baseline methods under such a scenario.

We then adopt the D4RL MuJoCo "-v2" expert-level datasets as source domain datasets. We allow the quality of the target domain dataset to be medium, medium-expert, or expert. We choose the morphology shift tasks and run all algorithms upon them for 1M gradient steps across 5 random seeds. The experimental results are shown in Table 7. It can be seen that OTDF outperforms other methods on **6** out of 12 tasks. OTDF achieves a total normalized score of **393.0**, while the second

Table 5: **Hyperparameter setup for OTDF and baseline methods.**

| Hyperparameter | Value |
|---|---|
| **Shared** | |
| Actor network | $(256, 256)$ |
| Critic network | $(256, 256)$ |
| Learning rate | $3 \times 10^{-4}$ |
| Optimizer | Adam (Kingma & Ba, 2015) |
| Discount factor | 0.99 |
| Nonlinearity | ReLU |
| Target update rate | $5 \times 10^{-3}$ |
| Source domain Batch size | 128 |
| Target domain Batch size | 128 |
| **IQL** | |
| Temperature coefficient | 0.2 |
| Maximum log std | 2 |
| Minimum log std | $-20$ |
| Inverse temperature parameter $\beta_{\text{IQL}}$ | 3.0 |
| Expectile parameter $\tau$ | 0.7 |
| **DARA** | |
| Temperature coefficient | 0.2 |
| Classifier network | $(256, 256)$ |
| Reward penalty coefficient $\lambda$ | 0.1 |
| **BOSA** | |
| Temperature coefficient | 0.2 |
| Maximum log std | 2 |
| Minimum log std | $-20$ |
| Policy regularization coefficient $\lambda_{\text{policy}}$ | 0.1 |
| Transition coefficient $\lambda_{\text{transition}}$ | 0.1 |
| Threshold parameter $\epsilon, \epsilon'$ | $\log(0.01)$ |
| Value wight $\omega$ | 0.1 |
| CVAE ensemble size of the dynamics model | 5 |
| **SRPO** | |
| Discriminator network | $(256, 256)$ |
| Data selection ratio | 0.5 |
| Reward coefficient $\lambda$ | $\{0.1, 0.3\}$ |
| **IGDF** | |
| Representation dimension | $\{16, 64\}$ |
| Contrastive encoder network | $(256, 256)$ |
| Encoder pretraining steps | 7000 |
| Importance coefficient | 1.0 |
| Data selection ratio $\xi\%$ | 75% |
| **OTDF** | |
| CVAE training steps | 10000 |
| CVAE learning rate | 0.001 |
| Number of sampled latent variables $M$ | 10 |
| Standard deviation of Gaussian distribution | $\sqrt{0.1}$ |
| Cost function | cosine |
| Data filtering ratio $\xi\%$ | 80% |
| Policy coefficient $\beta$ | $\{0.1, 0.5\}$ |

Table 6: **Performance comparison under the *kinematic* shift**. half = halfcheetah, hopp = hopper, walk = walker2d, m = medium, r = replay, e = expert. The target column denotes the offline dataset quality of the target domain. We report normalized scores and standard deviations in the *target domain* under varied dataset qualities of the source domain data (medium, medium-replay, medium-expert) and target domain data (medium, medium-expert, expert). The results are averaged over 5 seeds. We **bold** and highlight the best cell.

| Source | Target | IQL | DARA | BOSA | SRPO | IGDF | OTDF (ours) |
|---|---|---|---|---|---|---|---|
| half-m | medium | 12.3±1.2 | 10.6±1.2 | 8.3±1.2 | 16.8±4.2 | 23.6±5.7 | **40.2**±0.0 |
| half-m | medium-expert | 10.8±1.9 | **12.9**±2.8 | 8.7±1.3 | 10.3±2.7 | 9.8±2.4 | 10.1±4.0 |
| half-m | expert | 12.6±1.7 | 12.1±1.0 | 10.8±1.7 | 12.2±0.9 | **12.8**±0.7 | 8.7±2.0 |
| half-m-r | medium | 10.0±5.4 | 11.5±4.9 | 7.5±3.1 | 10.2±3.7 | 11.6±4.6 | **37.8**±2.1 |
| half-m-r | medium-expert | 6.5±3.1 | 9.2±4.7 | 6.6±1.7 | 9.5±1.8 | 8.6±2.3 | **9.7**±2.0 |
| half-m-r | expert | 13.6±1.4 | **14.8**±2.0 | 10.4±4.9 | **14.8**±2.2 | 13.9±2.2 | 7.2±1.4 |
| half-m-e | medium | 21.8±6.5 | 25.9±7.4 | 30.0±4.3 | 17.2±3.3 | 21.9±6.5 | **30.7**±9.6 |
| half-m-e | medium-expert | 7.6±1.4 | 9.5±4.2 | 6.8±2.9 | 9.6±2.4 | 8.9±3.3 | **10.9**±4.2 |
| half-m-e | expert | 9.1±2.4 | 10.4±1.3 | 4.9±3.2 | **11.2**±1.0 | 10.7±1.4 | 3.2±0.6 |
| hopp-m | medium | 58.7±8.4 | 43.9±15.2 | 12.3±6.6 | 65.4±1.5 | 65.3±1.4 | **65.6**±1.9 |
| hopp-m | medium-expert | **68.5**±12.4 | 55.4±16.9 | 15.6±10.8 | 43.9±30.8 | 51.1±18.5 | 55.4±25.1 |
| hopp-m | expert | 79.9±35.5 | 83.7±19.6 | 14.8±5.5 | 53.1±39.8 | **87.4**±25.4 | 35.0±19.4 |
| hopp-m-r | medium | 36.0±0.1 | **39.4**±7.2 | 3.2±2.6 | 36.1±0.2 | 35.9±2.4 | 35.5±12.2 |
| hopp-m-r | medium-expert | 36.1±0.1 | 34.1±3.6 | 4.4±2.8 | 36.0±0.1 | 36.1±0.1 | **47.5**±14.6 |
| hopp-m-r | expert | 36.0±0.1 | 36.1±0.2 | 3.7±2.5 | 36.1±0.1 | 36.1±0.3 | **49.9**±30.5 |
| hopp-m-e | medium | **66.0**±0.5 | 61.1±4.0 | 35.0±20.1 | 64.6±2.6 | 65.2±1.5 | 65.3±2.4 |
| hopp-m-e | medium-expert | 45.1±15.7 | 61.9±16.9 | 13.9±4.9 | 54.7±17.0 | **62.9**±15.6 | 38.6±15.9 |
| hopp-m-e | expert | 44.9±19.8 | **84.2**±21.1 | 12.0±4.3 | 57.6±40.6 | 52.8±39.7 | 29.9±11.3 |
| walk-m | medium | 34.3±9.8 | 35.2±22.5 | 14.3±11.2 | 39.0±6.7 | 41.9±11.2 | **49.6**±18.0 |
| walk-m | medium-expert | 30.2±12.5 | **51.9**±11.5 | 13.6±7.7 | 38.6±6.5 | 42.3±19.3 | 43.5±16.4 |
| walk-m | expert | 56.4±18.2 | 40.7±14.4 | 15.3±2.5 | 57.3±12.2 | **60.4**±17.5 | 46.7±13.6 |
| walk-m-r | medium | 11.5±7.1 | 12.5±4.3 | 1.9±2.1 | 14.3±3.1 | 22.2±5.2 | **49.7**±9.7 |
| walk-m-r | medium-expert | 9.7±3.8 | 11.2±5.0 | 4.6±3.0 | 4.2±5.1 | 7.6±4.9 | **55.9**±17.1 |
| walk-m-r | expert | 7.7±4.8 | 7.4±2.4 | 3.6±1.5 | 13.2±8.5 | 7.5±2.1 | **51.9**±7.9 |
| walk-m-e | medium | 41.8±8.8 | 38.1±14.4 | 21.4±8.3 | 36.9±4.3 | 41.2±13.0 | **44.6**±6.0 |
| walk-m-e | medium-expert | 22.2±8.7 | 23.6±8.1 | 15.9±4.1 | 23.2±7.9 | **28.1**±4.0 | 16.5±7.2 |
| walk-m-e | expert | 26.3±10.4 | 36.0±9.2 | 18.5±3.6 | 40.9±9.6 | **46.2**±19.4 | 42.4±9.1 |
| ant-m | medium | 50.0±5.6 | 42.3±7.6 | 20.9±2.6 | 50.5±6.7 | 54.5±1.3 | **55.4**±0.0 |
| ant-m | medium-expert | 57.8±7.2 | 54.1±3.8 | 31.7±7.0 | 54.9±1.3 | 54.5±4.6 | **60.7**±3.6 |
| ant-m | expert | 59.6±18.5 | 54.2±11.3 | 45.4±8.6 | 45.5±9.3 | 49.4±14.6 | **90.4**±4.8 |
| ant-m-r | medium | 43.7±4.6 | 42.0±5.4 | 19.0±1.8 | 45.3±5.1 | 41.4±5.0 | **52.8**±4.4 |
| ant-m-r | medium-expert | 36.5±5.9 | 36.0±6.7 | 19.1±1.6 | 36.2±6.6 | 37.2±4.7 | **54.2**±5.2 |
| ant-m-r | expert | 24.4±4.8 | 22.1±0.4 | 19.5±0.8 | 27.1±3.7 | 24.3±2.8 | **74.7**±10.5 |
| ant-m-e | medium | 49.5±4.1 | 44.7±4.3 | 19.0±8.0 | 41.3±8.1 | 41.8±8.8 | **50.2**±4.3 |
| ant-m-e | medium-expert | 37.2±2.0 | 33.3±7.0 | 6.4±2.5 | 32.8±8.0 | 41.5±4.9 | **48.8**±2.7 |
| ant-m-e | expert | 18.7±8.1 | 17.8±23.6 | 14.5±9.0 | 35.2±15.5 | 14.4±22.9 | **78.4**±12.2 |
| | Total Score | 1193.0 | 1219.8 | 513.5 | 1195.7 | 1271.0 | **1547.6** |

best approach, DARA, only achieves a total normalized score of 298.1. Despite that OTDF exhibits inferior performance on some tasks here, we would argue that *all hyperparameters adopted in OTDF are fixed*, i.e., $\xi\% = 80\%, \beta = 0.5$, without any hyperparameter tuning. Nevertheless, we tune hyperparameters for baseline methods. We strongly believe that the performance of OTDF can be further improved by carefully tuning the data selection ratio and the policy coefficient.

### E.3 IS IT NECESSARY TO WEIGHT SOURCE DOMAIN DATA?

In this part, we provide an ablation study to investigate whether we should adaptively weight the source domain data with $\exp(\widehat{d})$. To be specific, we consider a variant of OTDF that leverages the following objective function to update its state-action value function:

$$\mathcal{L}_Q = \mathbb{E}_{D_{\mathrm{tar}}}\left[(Q_\theta - \mathcal{T}Q_\theta)^2\right] + \mathbb{E}_{(s,a,s')\sim D_{\mathrm{src}}}\left[\mathbb{1}(d > d_{\xi\%})(Q_\theta - \mathcal{T}Q_\theta)^2\right]. \tag{30}$$

Table 7: **Performance comparison under the *morphology* shift tasks given expert-level source domain datasets**. half = halfcheetah, hopp = hopper, walk = walker2d, e = expert. All methods are run over 5 varied random seeds. We report normalized scores along with the corresponding standard deviations in the *target domain* given different qualities of target domain data (medium, medium-expert, expert). We **bold** and highlight the best cell.

| Source | Target | IQL | DARA | BOSA | SRPO | IGDF | OTDF (ours) |
|---|---|---|---|---|---|---|---|
| half-e | medium | 40.1±1.0 | **40.9**±1.5 | 40.5±1.5 | 40.8±1.0 | 39.4±2.6 | 38.6±1.4 |
| half-e | medium-expert | 22.5±3.6 | 27.9±0.6 | **28.7**±2.7 | 23.1±3.9 | 24.3±1.2 | 27.4±3.6 |
| half-e | expert | 7.9±1.5 | 8.6±0.5 | 8.0±0.7 | 6.3±0.8 | 7.4±1.3 | **8.8**±1.4 |
| hopp-e | medium | 9.5±2.3 | **11.4**±0.5 | 8.7±1.7 | 10.5±1.1 | 9.6±3.6 | 5.7±1.2 |
| hopp-e | medium-expert | 9.8±2.8 | 10.0±1.8 | 8.5±1.8 | 10.7±1.6 | **11.5**±0.4 | 6.5±1.4 |
| hopp-e | expert | 10.3±2.9 | 9.6±3.9 | 8.4±3.3 | **11.9**±0.3 | 10.2±2.8 | 5.7±4.0 |
| walk-e | medium | 36.7±4.6 | 36.9±4.5 | 6.1±5.1 | 36.7±7.7 | **38.6**±10.2 | 32.4±5.1 |
| walk-e | medium-expert | 20.6±7.2 | 29.2±9.0 | 4.5±2.9 | 21.9±5.6 | 30.1±5.9 | **34.8**±8.8 |
| walk-e | expert | 16.4±10.9 | 30.0±15.7 | 11.4±12.1 | 21.9±8.1 | 32.8±22.4 | **42.5**±17.1 |
| ant-e | medium | 31.0±9.0 | 39.6±2.5 | 33.0±3.5 | 33.5±4.1 | 35.8±8.1 | **40.4**±1.8 |
| ant-e | medium-expert | 28.1±4.2 | 37.8±9.5 | 47.3±13.8 | 37.4±10.4 | 36.1±3.8 | **61.4**±6.8 |
| ant-e | expert | 4.0±12.0 | 16.2±17.8 | 69.3±7.8 | 18.7±16.1 | 4.0±7.5 | **88.8**±6.6 |
| Total Score | | 236.9 | 298.1 | 274.4 | 273.4 | 279.8 | **393.0** |

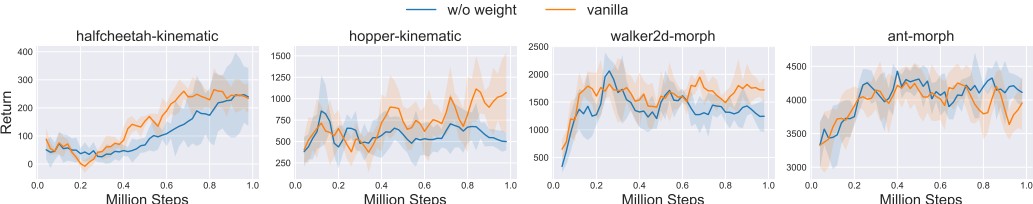

Figure 5: **Ablation study on the source domain data weight.** *-kinematic denotes tasks with kinematic shifts and *-morph means environments with morphology shifts. *w/o weight* refers to OTDF that excludes the component of source domain data weight, and *vanilla* denotes the vanilla OTDF algorithm. We report average returns in the target domain across 5 different random seeds and the shaded area captures the standard deviation.

Compared to Equation 9, the above objective function treats each filtered data with equal weights. Intuitively, this can be problematic since the data selection ratio $\xi\%$ is a constant, and bad transitions can still be included for training if no adaptive weighting mechanism (or regularization on the source domain data) is involved. Empirically, we conduct experiments on some selected tasks (two *kinematic* tasks and two *morphology* tasks) using the medium-replay source domain dataset and the expert target domain dataset. We present the experimental results in Figure 5. It is evident that the vanilla OTDF beats OTDF w/o weight on 3 out of 4 tasks, indicating that incorporating the weights $\exp(\widehat{d})$ is a better choice.

## F COMPUTE INFRASTRUCTURE

In Table 8, we list the compute infrastructure that we use to run all of the algorithms.

Table 8: **Compute infrastructure.**

| CPU | GPU | Memory |
|---|---|---|
| AMD EPYC 7452 | RTX3090×8 | 288GB |

