# OpenReview forum: "Cross-Domain Offline Policy Adaptation with Optimal Transport and Dataset Constraint"
_ICLR.cc/2025/Conference — ICLR 2025 Poster_

### Official Review · Reviewer_vhAr · 2024-11-02

**Soundness:** 4
**Presentation:** 3
**Contribution:** 3
**Rating:** 8
**Confidence:** 2

**Summary:**

This work proposes a new cross-domain offline RL method OTDF, which can leverage data from source domain to help the learning of the target domain. This work proposes that the core of the cross-domain offline RL is the dynamics mismatch between source and target domain, and introduces a threshold and a weight to only leverage those data close to the target domain in the source domain.

**Strengths:**

- This work proposes an informative proof and well-designed method. Also, this work can be a well baseline for follow-up works.

**Weaknesses:**

The $\mu_{t,t'}$ is not defined in Eq. 1.

**Questions:**

Can a more complex kinematic shift be proposed?

---

> ### Author Response · Authors · 2024-11-20
> **Author Responses to Reviewer vhAr**
>
> We thank the reviewer for commenting that our work proposes an informative proof and well-designed method and that our method can serve as a well baseline for follow-up works. Please check our clarifications to the concerns below.
>
> **Concern 1: $\mu_{t,t^\prime}$ is not defined**
>
> Thanks for the question. $\mu\in \mathbb{R}^{n\times n^\prime}$ is the coupling matrix, and $\mu_{t,t^\prime}$ denotes its $t$-th row, $t^\prime$-th column element, $t=1,\ldots,n, t^\prime=1,\ldots,n^\prime$, which corresponds to the coupling with respect to $x\_t, y\_{t^\prime}$. We have made it clearer in the revision.
>
> **Concern 2: Can a more complex kinematic shift be proposed?**
>
> Yes, it is feasible. The kinematic shift can occur at different parts of the robot.
>
> - for halfcheetah task, we modify the rotation angle of the joint on the thigh of the robot’s back leg in the submission. Furthermore, one can modify the rotation range of the foot joint to fulfill kinematic mismatch
> - for hopper task, one can modify the rotation range of either the head joint or the foot joint (we already modified the rotation angle of both the head joint and the foot joint in our paper)
> - for walker2d task, we modify the rotation angle of the foot joint on the robot’s right leg in our submission. One can also modify the rotation range of the thigh joint.
> - for ant task, we modify the rotation angles of the joints on the hip of two legs in our original submission. We can also modify the rotation range of the ankle joint
>
> Note that the joints can be broken simultaneously, and it depends on the user to design the kinematic shift (e.g., the broken joints occur only in one joint or multiple joints).
>
> To examine the effectiveness of OTDF under possibly more complex kinematic shifts, we consider two domains (walker2d, ant) and modify their XML files. We only evaluate on two domains because we need to train the agent in the modified environment to ensure that the agent can get meaningful performance and modify the XML files if not. We also need to gather offline datasets with different qualities. All these consume a lot of time. For the walker2d task, we modify the rotation range of the thigh joint in the target domain to be 0.2 times of that in the source domain:
> ```
> <joint axis="0 -1 0" name="thigh_joint" pos="0 0 1.05" range="-30 0" type="hinge"/>
> <joint axis="0 -1 0" name="thigh_left_joint" pos="0 0 1.05" range="-30 0" type="hinge"/>
> ```
>
> For ant task, we modify the rotation range of the ankle joint of all legs from [30, 70] to [30, 38]:
> ```
> <joint axis="-1 1 0" name="ankle_1" pos="0.0 0.0 0.0" range="30 38" type="hinge"/>
> <joint axis="1 1 0" name="ankle_2" pos="0.0 0.0 0.0" range="-38 -30" type="hinge"/>
> <joint axis="-1 1 0" name="ankle_3" pos="0.0 0.0 0.0" range="-38 -30" type="hinge"/>
> <joint axis="1 1 0" name="ankle_4" pos="0.0 0.0 0.0" range="30 38" type="hinge"/>
> ```
>
> We then first train the SAC agent in the above new kinematic tasks and gather medium, medium-expert, and expert offline datasets following D4RL. All target domain datasets still have a limited budget of 5000 transitions. Due to the limited rebuttal period, we only run OTDF and IQL* (with both domain data) on some modified kinematic tasks (medium/medium-replay/medium-expert source domain dataset and medium/medium-expert target domain dataset). We use the default hyperparameters of OTDF and summarize the results below. All results are averaged across 5 different random seeds, and we report the normalized scores on the target domain.
>
> | Source | Target | IQL* | OTDF (ours) |
> | ---- | ---- | :---: | :---: |
> | walk-m | medium | 64.7$\pm$4.1 | **67.0**$\pm$0.0 |
> | walk-m | medium-expert | **64.6**$\pm$5.8 | 60.6$\pm$11.1 |
> | walk-m-r | medium | 51.9$\pm$8.1 | **54.3**$\pm$0.0 |
> | walk-m-r | medium-expert | **52.2**$\pm$12.9 | 51.7$\pm$16.0 |
> | walk-m-e | medium | 97.6$\pm$16.4 | **98.1**$\pm$16.8 |
> | walk-m-e | medium-expert | **95.5**$\pm$6.6 | 93.7$\pm$3.5 |
> | ant-m | medium | 59.8$\pm$5.7 | **81.0**$\pm$4.8 |
> | ant-m | medium-expert | 43.2$\pm$2.0 | **74.6**$\pm$11.5 |
> | ant-m-r | medium | 50.1$\pm$4.4 | **78.4**$\pm$10.4 |
> | ant-m-r | medium-expert | 41.9$\pm$1.0 | **62.4**$\pm$6.4 |
> | ant-m-e | medium | 73.1$\pm$1.6 | **83.4**$\pm$5.7 |
> | ant-m-e | medium-expert | 71.3$\pm$2.6 | **80.8**$\pm$14.3 |
> | Total | | 765.8 | **886.0** |
>
> Table 1. Results on new kinematic tasks. walk=walker2d, m=medium, r=replay, e=expert. We report the mean normalized score with the standard deviations.
>
> It can be seen that OTDF still outperforms IQL* on most of the tasks, further demonstrating the effectiveness of our method. One can also modify the kinematic tasks by simulating broken joints at multiple joints (e.g., simultaneously broken ankle joints and hip joints at the ant task) to construct more diverse and complex kinematic shifts and run OTDF on them.
>
> Hopefully, these can resolve the concerns. If there is still something unclear, please let us know!

---

> > ### Author Response · Authors · 2024-11-24
> > **Following up with Reviewer vhAr**
> >
> > Dear Reviewer vhAr, thank you very much for the helpful review and positive rating of our work. It would be great if you could give us some comments on our rebuttal, and kindly check our revision.

---

> > > ### Author Response · Authors · 2024-11-28
> > >
> > > Dear Reviewer vhAr, thank you for your constructive review! We hope that our rebuttal and the revised manuscript can address your concerns. We would appreciate it if you could give us some feedback. Please let us know if there is still anything unclear!

---

### Official Review · Reviewer_6ito · 2024-11-04

**Soundness:** 2
**Presentation:** 2
**Contribution:** 3
**Rating:** 6
**Confidence:** 3

**Summary:**

This paper addresses the challenge of cross-domain offline reinforcement learning (RL) where the goal is to leverage a source domain dataset to improve policy learning in a target domain with limited data. This paper identifies that directly merging data from different domains can lead to performance degradation due to dynamics mismatches. To mitigate this, they propose a method called Optimal Transport Data Filtering (OTDF), which aligns transitions from source and target domains using optimal transport, selectively shares source domain samples, and introduces a dataset regularization term to keep the learned policy within the scope of the target domain dataset. The effectiveness of OTDF is evaluated across various dynamics shift conditions with limited target domain data, demonstrating superior performance over strong baselines.

**Strengths:**

1. This paper introduces a novel approach to cross-domain offline RL by combining optimal transport with dataset regularization, which is a unique combination not commonly seen in the literature.
2. The authors provide a solid theoretical foundation for their method, characterizing the performance bound of a policy and motivating their approach with theoretical insights

**Weaknesses:**

1. In previous studies, DARA, BOSA, and IGDF all utilize source and target domain data of similar quality, implying that the state space distributions of the datasets were closely aligned. This alignment ensures, to some extent, that there is a certain overlap between the source and target domain data. Your setting differs from theirs, which naturally raises the question of whether data filtering or policy regularization played the critical role. As the results in Figure 3 indicate, the introduction of policy regularization has a highly significant impact on the performance of the algorithm. However, in previous methods such as SRPO and IGDF, constraints related to this were not designed, which may account for the significant performance improvement observed in OTDF.

2. In the paper "Cross-domain imitation learning via optimal transport, ICLR 2022," some methods for cross-domain transfer based on optimal transport have already been adopted. What do you think are the differences between your method and the previous methods in terms of implementation?

**Questions:**

1. Table 1 is too wide and not pleasing enough. Recommend using \resizebox {\linewidth} {!} to fix it.
2. Why is the performance of IQL not assessed when using only source domain data?
3. The paper does not provide the impact of using OT for data filtering or subsequent policy regularization on the performance bound of OTDF. A sub-optimal gap is expected to be given.

---

> ### Author Response · Authors · 2024-11-20
> **Author Responses to Reviewer 6ito (part 1)**
>
> We thank the reviewer for the constructive comments. We provide point-to-point clarification to the concerns of the reviewer. If we are able to resolve some concerns, we hope that the reviewer will be willing to raise the score.
>
> **Concern 1: on the experiment setting**
>
> Thanks for the question. We use source and target domain data of distinct qualities because (a) we can better capture the generality and effectiveness of the cross-domain offline RL methods and there is no definite rule that one can only conduct cross-domain offline policy adaptation for source domain offline datasets and target domain datasets with the same quality level; (b) in many real-world applications, we often cannot provide exactly the same performance level offline dataset for both domains.
>
> We agree that there is a certain overlap between the source and target domain data **to some extent** when they are of similar quality (it is hard to quantify that though). Nevertheless, we respectfully argue that they can also have some overlap even when they are of different qualities since they are still striving towards the same goal. For example, the robot aims at walking forward in the walker2d task. The expert-level walker2d dataset with kinematic shift can also share some overlap with the medium-level source domain dataset (e.g., some walking movements and patterns). Furthermore, when one adopts a medium-level target domain dataset, it can **at least** share overlap with medium/medium-replay/medium-expert source domain datasets since they both contain medium-level data. It also applies to target domain datasets with medium-expert quality and expert quality. Hence, we believe that it is reasonable to adopt a setting like this. Moreover, OTDF still significantly outperforms baseline methods like IGDF and SRPO across various kinds of dynamics shifts when the source domain dataset and target domain dataset have the same performance level (e.g., medium-level source domain data and medium-level target domain data), as depicted in Table 1, Table 2 and Table 6.
>
> We further clarify that both data filtering based on optimal transport and policy regularization indeed play the critical role. We reiterate that our method is fully motivated by the theoretical results in Theorem 3.1 and the experimental setting where we only have a limited budget of target domain data. Data filtering effectively controls the dynamics mismatch term (c) in Theorem 3.1 by keeping source domain data that have similar transition dynamics as the target domain dynamics (as discussed in Lines 160-165). Optimal transport is a principled approach for comparing two distributions and is ideal to be applied for reliable data filtering due to its training-free nature, given that we only have limited target domain data.  Furthermore, the dataset constraint objective better controls the target policy deviation term (b) in Theorem 3.1 and ensures that the learned policy would not get biased towards the source domain dataset distribution considering that the target domain dataset size is limited. The parameter study in Figure 2 and Figure 3 show that excluding either data filtering ($\xi=100$) or dataset constraint ($\beta=0$) often incurs a performance drop, **indicating the necessity and effectiveness of both components to OTDF**. The influence of data filtering can also be large for many tasks. We agree that the introduction of the dataset constraint term is important to OTDF and previous methods like SRPO and IGDF do not include it, but this component is still our novel contribution.
>
> **Concern 2: differences between our work and some prior works**
>
> Thanks for recommending GWIL [1], which we have actually included in our original submission. Despite that we both utilize optimal transport in the cross-domain setting, our method differs from GWIL [1] in the following aspects:
>
> - the motivations are different. GWIL [1] solves the cross-domain imitation learning problem from a new perspective by directly formalizing it as an optimal transport problem. Our method, instead, utilizes optimal transport (OT) due to (a) OT is a principled and widely used approach for comparing two distributions; (b) we only have a very limited budget of target domain data, and OT can work given such low data regime thanks to its training-free nature.
> - the experimental settings are different. GWIL focuses on the cross-domain imitation learning scenario, where expert demonstrations are needed. Our OTDF addresses the cross-domain offline RL problems with different performance level offline source domain or target domain datasets (no expert demonstration is required).
> - the core methods are different. GWIL leverages the optimal transport for constructing pseudo-rewards while OTDF utilizes the optimal transport for data filtering (i.e., keeping source domain data that have similar transition dynamics as the target domain dynamics)
>
> [1] Cross-domain imitation learning via optimal transport

---

> > ### Author Response · Authors · 2024-11-20
> > **Author Responses to Reviewer 6ito (part 2)**
> >
> > **Concern 3: Table 1 is too wide and not pleasing enough**
> >
> > Thanks for the comment. We have modified all tables in our manuscript to ensure that they look pleasing.
> >
> > **Concern 4: Why is the performance of IQL not assessed when using only source domain data?**
> >
> > We do not report the performance of IQL with only source domain data since we care more about the agent's performance in the target domain. Considering the fact that there exists a dynamics gap between the source domain and the target domain, directly deploying IQL trained with only source domain data may incur inferior performance in the target domain. We hence choose to report IQL\*, which trains IQL using both the source domain data and target domain data. Since the reviewer asks, we conduct experiments of IQL (source only) on some tasks with kinematic shifts and morphology shifts. We report the average normalized scores across 5 seeds and summarize the results below. Note that the performance of IQL (source only) remains identical across different target domain dataset qualities since it is trained using only source domain data. We find that IQL with only source domain data often incurs worse performance than OTDF, and underperforms IQL\*, indicating that it may not be a meaningful baseline.
> >
> > | Source | Target | IQL (source only) | IQL* | OTDF |
> > | ---- | ---- | :---: | :---: | :---: |
> > | half-m | medium | 14.7$\pm$2.0 | 12.3$\pm$1.2 | **40.2**$\pm$0.0 |
> > | half-m | medium-expert | **14.7**$\pm$2.0 | 10.8$\pm$1.9 | 10.1$\pm$4.0 |
> > | half-m-r | medium | 14.5$\pm$2.1 | 10.0$\pm$5.4 | **37.8**$\pm$2.1 |
> > | half-m-r | medium-expert | **14.5**$\pm$2.1 | 6.5$\pm$3.1 | 9.7$\pm$2.0 |
> > | half-m-e | medium | 13.1$\pm$3.4 | 21.8$\pm$6.5 | **30.7**$\pm$9.6 |
> > | half-m-e | medium-expert | **13.1**$\pm$3.4 | 7.6$\pm$1.4 | 10.9$\pm$4.2 |
> > | hopp-m | medium | 31.5$\pm$9.2 | 58.7$\pm$8.4 | **65.6**$\pm$1.9 |
> > | hopp-m | medium-expert | 31.5$\pm$9.2 | **68.5**$\pm$12.4 | 55.4$\pm$25.1 |
> > | hopp-m-r | medium | 21.0$\pm$15.7 | **36.0**$\pm$0.1 | 35.5$\pm$12.2 |
> > | hopp-m-r | medium-expert | 21.0$\pm$15.7 | 36.1$\pm$0.1 | **47.5**$\pm$14.6 |
> > | hopp-m-e | medium | 8.7$\pm$6.8 | **66.0**$\pm$0.5 | 65.3$\pm$2.4 |
> > | hopp-m-e | medium-expert | 8.7$\pm$6.8 | **45.1**$\pm$15.7 | 38.6$\pm$15.9 |
> > | walk-m | medium | 40.0$\pm$15.6 | 34.3$\pm$9.8 | **49.6**$\pm$18.0 |
> > | walk-m | medium-expert | 40.0$\pm$15.6 | 30.2$\pm$12.5 | **43.5**$\pm$16.4 |
> > | walk-m-r | medium | 12.5$\pm$5.1 | 11.5$\pm$7.1 | **49.7**$\pm$9.7 |
> > | walk-m-r | medium-expert | 12.5$\pm$5.1 | 9.7$\pm$3.8 | **55.9**$\pm$17.1 |
> > | walk-m-e | medium | 24.1$\pm$12.2 | 41.8$\pm$8.8 | **44.6**$\pm$6.0 |
> > | walk-m-e | medium-expert | **24.1**$\pm$12.2 | 22.2$\pm$8.7 | 16.5$\pm$7.2 |
> > | ant-m | medium | 19.5$\pm$4.2 | 50.0$\pm$5.6 | **55.4**$\pm$0.0 |
> > | ant-m | medium-expert | 19.5$\pm$4.2 | 57.8$\pm$7.2 | **60.7**$\pm$3.6 |
> > | ant-m-r | medium | 22.3$\pm$0.5 | 43.7$\pm$4.6 | **52.8**$\pm$4.4 |
> > | ant-m-r | medium-expert | 22.3$\pm$0.5 | 36.5$\pm$5.9 | **54.2**$\pm$5.2 |
> > | ant-m-e | medium | 16.4$\pm$2.5 | 49.5$\pm$4.1 | **50.2**$\pm$4.3 |
> > | ant-m-e | medium-expert | 16.4$\pm$2.5 | 37.2$\pm$2.0 | **48.8**$\pm$2.7 |
> > | Total | | 476.6 | 803.8 | **1029.2** |
> >
> > Table 1. Results on kinematic tasks. IQL* denotes that IQL is trained with both the source domain dataset and target domain dataset. half=halfcheetah, hopp=hopper, walk=walker2d, m=medium, r=replay, e=expert. We report the mean normalized scores along with their standard deviations.

---

> > > ### Author Response · Authors · 2024-11-20
> > > **Author Responses to Reviewer 6ito (part 3)**
> > >
> > > | Source | Target | IQL (source only) | IQL* | OTDF |
> > > | ---- | ---- | :---: | :---: | :---: |
> > > | half-m | medium | 5.6$\pm$0.8 | 30.0$\pm$1.6 | **39.1**$\pm$2.3 |
> > > | half-m | medium-expert | 5.6$\pm$0.8 | 31.8$\pm$1.1 | **35.6**$\pm$0.7 |
> > > | half-m-r | medium | 6.5$\pm$0.4 | 30.8$\pm$4.4 | **40.0**$\pm$1.2 |
> > > | half-m-r | medium-expert | 6.5$\pm$0.4 | 12.9$\pm$2.2 | **34.4**$\pm$0.7 |
> > > | half-m-e | medium | 5.5$\pm$0.5 | **41.5**$\pm$0.1 | 41.4$\pm$0.3 |
> > > | half-m-e | medium-expert | 5.5$\pm$0.5 | 25.8$\pm$2.0 | **35.1**$\pm$0.6 |
> > > | hopp-m | medium | 13.2$\pm$0.1 | **13.5**$\pm$0.2 | 11.0$\pm$0.9 |
> > > | hopp-m | medium-expert | 13.2$\pm$0.1 | **13.4**$\pm$0.1 | 12.6$\pm$0.8 |
> > > | hopp-m-r | medium | **12.0**$\pm$0.2 | 10.8$\pm$1.1 | 8.7$\pm$2.8 |
> > > | hopp-m-r | medium-expert | **12.0**$\pm$0.2 | 11.6$\pm$1.6 | 9.7$\pm$2.7 |
> > > | hopp-m-e | medium | 10.3$\pm$2.8 | **12.6**$\pm$1.4 | 7.9$\pm$3.2 |
> > > | hopp-m-e | medium-expert | 10.3$\pm$2.8 | **14.1**$\pm$1.3 | 9.6$\pm$3.5 |
> > > | walk-m | medium | 13.2$\pm$1.7 | 23.0$\pm$4.7 | **50.5**$\pm$5.8 |
> > > | walk-m | medium-expert | 13.2$\pm$1.7 | 21.5$\pm$8.6 | **44.3**$\pm$23.8 |
> > > | walk-m-r | medium | 10.7$\pm$1.6 | 11.3$\pm$3.0 | **37.4**$\pm$5.1 |
> > > | walk-m-r | medium-expert | 10.7$\pm$1.6 | 7.0$\pm$1.5 | **33.8**$\pm$6.9 |
> > > | walk-m-e | medium | 13.6$\pm$6.7 | 24.1$\pm$7.4 | **49.9**$\pm$4.6 |
> > > | walk-m-e | medium-expert | 13.6$\pm$6.7 | 27.0$\pm$5.5 | **40.5**$\pm$11.0 |
> > > | ant-m | medium | 32.7$\pm$1.5 | 38.7$\pm$3.8 | **39.4**$\pm$1.7 |
> > > | ant-m | medium-expert | 32.7$\pm$1.5 | 47.0$\pm$5.1 | **58.3**$\pm$8.9 |
> > > | ant-m-r | medium | 30.3$\pm$2.8 | 38.2$\pm$2.9 | **41.2**$\pm$0.9 |
> > > | ant-m-r | medium-expert | 30.3$\pm$2.8 | 38.1$\pm$3.5 | **50.8**$\pm$4.5 |
> > > | ant-m-e | medium | 30.6$\pm$3.4 | 32.9$\pm$5.1 | **39.9**$\pm$2.9 |
> > > | ant-m-e | medium-expert | 30.6$\pm$3.4 | 35.7$\pm$3.9 | **65.7**$\pm$4.5 |
> > > | Total | | 368.4 | 593.3 | **836.8** |
> > >
> > > Table 2. Results on morphology tasks. We report the mean normalized scores along with their standard deviations.
> > >
> > > **Concern 5: the impact of using OT for data filtering or subsequent policy regularization on the performance bound of OTDF**
> > >
> > > We respectfully clarify that both data filtering with optimal transport and policy regularization are motivated by the theoretical results in Theorem 3.1 as clarified in our responses to **Concern 1** above. Their impact on the performance bound can actually be qualitatively seen in Theorem 3.1, i.e., when using optimal transport for data filtering, we keep source domain data that is close to the target domain data, which indicates that we actually modify the empirical distribution of the source domain dataset $\widehat{\mathcal{M}}\_{\rm src}$ to become closer to the target domain, i.e., term (c) $D\_{\rm TV}(P\_{\mathcal{M}\_{\rm tar}}\| P\_{\widehat{\mathcal{M}}\_{\rm src}})$ in Theorem 3.1 can be better controlled. Policy regularization helps control term (b) $D\_{\rm TV}(\pi\_{D\_{\rm tar}}\|\pi)$. A rigorous and quantitative sub-optimality gap after introducing data filtering and policy regularization can be difficult to derive (to the best of our knowledge, none of the previous cross-domain offline RL papers successfully build such sub-optimality gaps based on their method yet) and may need some strong assumptions. These can yield gaps between theoretical analysis and practical application. We hence are sorry for not being able to provide the expected sub-optimality gaps but would be eager to explore that in future works.
> > >
> > > Hopefully, these can resolve the concerns. If there is still something unclear, please let us know!

---

> > > > ### Author Response · Authors · 2024-11-24
> > > > **Following up with Reviewer 6ito**
> > > >
> > > > Dear Reviewer 6ito, thanks for your time and efforts in making our paper better. Since the deadline of Author-Reviewer discussion period draws near, we wonder if you can kindly check our rebuttal and see if our responses mitigate your concerns. We would appreciate it if you could give us some feedback, and we are ready to have further discussions with the reviewer if there is anything unclear.

---

> > > > > ### Author Response · Authors · 2024-11-28
> > > > >
> > > > > Dear Reviewer 6ito, thank you for your helpful review! We would like to double-check if our response can address your concerns. Please do not hesitate to let us know if you still have any concerns or questions. We would appreciate it if the reviewer could re-evaluate our work based on the revised manuscript and the attached rebuttal. We are looking forward to your kind reply!

---

> > > > ### Comment · Reviewer_6ito · 2024-12-03
> > > >
> > > > Thank you for your response! The concerns have essentially been addressed, and as a result, we have raised the score to 6.

---

> > > > > ### Author Response · Authors · 2024-12-03
> > > > > **Thanks for raising the score!**
> > > > >
> > > > > We are glad that the reviewer's concerns are addressed. We thank the reviewer for raising the score to 6! Thanks for your time and efforts in making our paper better.

---

### Official Review · Reviewer_i1Jo · 2024-11-04

**Soundness:** 3
**Presentation:** 3
**Contribution:** 3
**Rating:** 6
**Confidence:** 4

**Summary:**

This paper proposes a novel method named Optimal Transport Data Filtering (OTDF) for cross-domain offline policy adaptation. Previous works still need a comparatively large target domain dataset to learn domain classifiers or filtering via mutual information. In this paper, the authors provide a performance bound to identify the factors of the performance deviation. This paper proposes to use optimal transport to align data distributions and a regularization term to ensure the learned policy remains within the target domain’s span. The empirical study shows that OTDF outperforms existing methods in cross-domain offline policy adaptation.

**Strengths:**

- The paper introduces a theoretical analysis of the performance bound in cross-domain offline policy adaptation, providing a clear foundation for the proposed method.
- The optimal transport-based data filtering method is motivated and efficient to compute the Wasserstein distance between source and target domains and filter the source data accordingly.
- The empirical results across various environments and types of dynamics shifts demonstrate that OTDF significantly outperforms baseline methods based on the IQL.

**Weaknesses:**

- The proposed practical implementation of OTDF still needs to learn a conditional variational auto-encoder to estimate the behavior polic which may introduce additional computational overhead.
- The paper argues that the proposed method can use a small target domain dataset however the detailed explanation of theoretical and empirical analysis is still limited.
- Ablation study on the proposed method is limited and the comparison with other dynamic model-based methods or adaptive transfer learning approaches is not comprehensive.

**Questions:**

- Can you provide more insights on using the small amount of target domain data in the proposed method? Can you provide a more detailed explanation from the theoretical and empirical sides?
- Can you explain more about the CVAE you used in the proposed method? Why use the CVAE to estimate the behavior policy? How does the CVAE affect the performance of the proposed method?
- Can you discuss and cite more related works on cross-domain policy adaptation such as [1], [2], and [3], and how the proposed method compares to these methods?
- Can you explain why use the different performance level target domain dataset from the source domain dataset in the empirical study? For example, why use the medium dataset in the source domain combined with the expert dataset in the target domain? Is it a reasonable setting for cross-domain policy adaptation?
- Do you have any results for other offline RL backbones like CQL, BCQ, or BEAR? How does the proposed method compare to these methods in cross-domain offline policy adaptation?

[1] When to trust your simulator: Dynamics-aware hybrid offline-and-online reinforcement learning

[2] Off-Dynamics Reinforcement Learning via Domain Adaptation and Reward Augmented Imitation

[3] Return Augmented Decision Transformer for Off-Dynamics Reinforcement Learning

---

> ### Author Response · Authors · 2024-11-20
> **Author Responses to Reviewer i1Jo (part 1)**
>
> We thank the reviewer for the insightful review. We appreciate that the reviewer acknowledges that our work provides a clear foundation for the proposed method and that OTDF significantly outperforms baseline methods. Please find our clarifications to the concerns below
>
> **Concern 1: on the computational overhead**
>
> We clarify that the CVAE is actually pre-trained before the policy learning stage of OTDF (as stated in Line 281). This indicates that no training costs of CVAE will be included during the training of OTDF. Meanwhile, the computation cost of training CVAE is also tolerable since we only train CVAE for 10000 steps (please see Table 5 for hyperparameter setup). Despite that measuring the log probabilities of the action lying in the span of the target domain dataset can still require some extra computational overhead. We respectfully argue that there is no free lunch. Many cross-domain offline RL methods also suffer from the same issue, e.g., DARA [1] needs to train domain classifiers and construct reward penalties, and IGDF [2] includes the contrastive learning objective for data filtering. These can all introduce additional computational overhead. Based on our experiments, we find that OTDF enjoys a runtime similar to those baseline algorithms
>
> **Concern 2: insights on using the small amount of target domain data in OTDF**
>
> The insights behind the small amount of target domain data lie in the following aspects:
> - under many scenarios, the target domain data is hard to acquire. It can be expensive to gather data in the target domain (e.g., real-world robotics data). It is somewhat unrealistic to train policies using a large amount of target domain data (e.g., 1e5). Nevertheless, a small amount of target domain data (e.g., 5000) is more acceptable and practicable
> - human beings are able to quickly adapt to the downstream tasks with only a small amount of data. For example, Alice used to play tennis without any exposure to other ball sports (Line 33 of the main text), and she never played badminton before. However, based on her experience in tennis, she can play badminton after a few rounds of play. We believe such ability is also expected in decision-making agents. Hence, we only provide a limited budget of target domain data to realize efficient offline policy adaptation
> - existing cross-domain offline RL papers still rely on a large amount of data (e.g., papers like DARA/IGDF/BOSA rely on 10\% D4RL data, which can still result in 1e5 data eventually, which is indeed a comparatively large amount of data). We note that this can downgrade the necessity of a source domain dataset since one can get satisfying performance on them with existing strong offline RL methods. This can be verified in Table 1 in the BOSA paper [3], and Table 1 in the IGDF paper, where offline RL methods like IQL, CQL, and SPOT can achieve quite strong performance with 10\% D4RL data. In contrast, we find that these offline RL methods typically fail under the single-domain setting when only 5000 data is available
>
> We are sorry that a strict and formal theoretical analysis of a small amount of data can be difficult. To the best of our knowledge, there are no suitable mathematical tools for connecting the performance bound of the policy and the dataset size in a general manner. There exist some works that try to do so, e.g., [4], but they often rely on some assumptions that deviate far from the practical applications
>
> **Concern 3: more explanations on the CVAE**
>
> We use CVAE to fulfill the dataset constraint, i.e., using CVAE to estimate the behavior policy of the target domain dataset and maximize the log probability of the current policy lying in the span of the target domain dataset. This can intuitively prevent the learned policy from getting biased towards the source domain data, which is vital since we focus exactly on the agent's performance in the target domain. Theoretically, incorporating the dataset constraint term can better control term (b) in Theorem 3.1. Please refer to more details in Lines 216-233 of the main text.
>
> We use CVAE to model the behavior policy due to the fact that (a) CVAE is a very popular generative model with good theoretical foundations, which can generate in-distribution samples; (b) CVAE is widely adopted and proven to be effective in the offline RL community to estimate the behavior policy, e.g., BCQ [5], SPOT [6], PLAS [7], etc. (c) Using CVAE can incur better performance. It turns out that CVAE is very important to OTDF. In Figure 3 of the main text, we conduct parameter study on the policy regularization coefficient $\beta$, which determines the strengths of the dataset constraint, i.e., a larger $\beta$ indicates a stronger constraint and vice versa. Setting $\beta=0$ usually leads to a significant performance drop. The impact of $\beta$ can be large on some tasks, indicating that CVAE is undoubtedly a critical component of OTDF.

---

> > ### Author Response · Authors · 2024-11-20
> > **Author Responses to Reviewer i1Jo (part 2)**
> >
> > **Concern 4: Ablation study and baseline comparison is not comprehensive**
> >
> > We respectfully argue that our ablation study is sufficient, where we show the effectiveness of data filtering in Figure 2 ($\xi=100$ indicates no data filtering, which usually incurs performance drop), and the effectiveness of dataset constraint term in Figure 3 ($\beta=0$ means no dataset constraint term, which often results in performance drop). We also provide the ablation study on the source domain data weight in Figure 5, which shows its necessity. All of these ablation studies are conducted on 4 tasks (2 kinematic tasks and 2 morphology tasks). Please let us know if the reviewer wants more environments to be covered in the ablation study, and we will add them to the final version.
> >
> > Our comparison against the baseline methods is also comprehensive. We would like to emphasize that we focus on the cross-domain offline policy adaptation scenario, where **both the source domain and the target domain are offline**, and no online interactions with either the source domain or the target domain are allowed. It indicates that many of the off-dynamics RL papers in the literature are not suitable or applicable to serve as baselines (e.g., DARC, H2O [8], DARAIL [9]). We have tried our best to include some representative and strong methods for comparison, e.g., DARA and BOSA. Meanwhile, IGDF is a very recently published cross-domain offline RL paper in ICML 2024 and achieves strong performance on numerous dynamics shift scenarios. We hence believe that our selected baselines are strong, relevant, and comprehensive.
> >
> > **Concern 5: discuss and cite more related works on cross-domain policy adaptation**
> >
> > Thanks for recommending these works. We have cited H2O [8] in our original submission. [9, 10] seem to be available online after the submission deadline of this venue. We find them quite relevant and have now included them in the revision. H2O [8] realizes policy adaptation via adaptively penalizing the Q-function learning on simulated state-action pairs with large dynamics gaps. It focuses on the setting of online-offline cross-domain RL, i.e., the source domain is online while the target domain is offline. DARAIL [9] first trains the DARC method using both source domain environment and target domain environment, and then transfers the policy’s behavior from the source to the target domain through imitation learning from observation. It focuses on the setting where both the source domain and the target domain are online. RADT [10] addresses the off-dynamics scenario from the return-conditioned supervised learning (RCSL) perspective, where it augments the return in
> > the source domain by aligning its distribution with that in the target domain. It aims at the cross-domain offline RL scenario.
> >
> > The differences between OTDF and these methods can be summarized as below:
> >
> > - the experimental setting differs. OTDF focuses on offline policy adaptation where both the source domain and the target domain are offline. RADT also studies this setting, while H2O focuses on an online source domain and an offline target domain, and DARAIL requires both an online source domain and an online target domain.
> > - DARAIL requires a comparatively large amount of online interactions with the target domain. For H2O and RADT, they both typically require a large amount of target domain offline data (either using 100\% D4RL data or 10\% D4RL data, corresponding to at least 100,000 target domain data). Instead, OTDF conducts experiments with very limited target domain data (5000 transitions)
> > - OTDF leverages optimal transport for data filtering and dataset constraint to avoid the learned policy from getting biased towards the source domain dataset distribution. These techniques are not observed in methods like DARAIL.

---

> > > ### Author Response · Authors · 2024-11-20
> > > **Author Responses to Reviewer i1Jo (part 3)**
> > >
> > > **Concern 6: why use the different performance level target domain dataset from the source domain dataset in the empirical study**
> > >
> > > We respectfully argue that this is a reasonable setting for cross-domain policy adaptation because:
> > >
> > > - there is no definite rule that one can only conduct cross-domain offline policy adaptation for source domain offline datasets and target domain datasets with the same quality level. We note that even medium-level source domain dataset and medium-level target domain dataset can have large discrepancies. For example, the medium-level halfcheetah-kinematic dataset has an average return of about 2700 (please check Table 4 of our manuscript for details), while the average return of D4RL halfcheetah-medium-v2 is about 4800, indicating that their behaviors and patterns can be distinct. Using different performance-level target domain datasets from the source domain dataset can better capture the generality and effectiveness of the cross-domain offline RL methods
> > > - in many real-world applications, we often cannot provide exactly the same performance level offline dataset for both domains. For example, we may gather medium-level offline datasets in version A of a game (i.e., the source domain) with some medium-level RL policy. The company decides to modify some features of this game in the new season and constructs version B of this game (i.e., the target domain). The company hires some expert human players to play the new version of this game and log their play data. The logged human data is of expert quality but limited size. It is infeasible to train a strong AI simply with a limited budget of data. Then, we naturally would leverage medium-level source domain data to realize efficient offline policy adaptation. This example also applies to real-world robotics datasets.
> > >
> > > **Concern 7: results with other offline RL backbones**
> > >
> > > Interesting question! In our work, we use IQL as the backbone because (a) other baseline methods like IGDF, and DARA use IQL as the base algorithm, we then also use IQL to ensure a fair comparison; (b) IQL naturally satisfies the in-distribution constraint that does not query any OOD samples, which is a nice property to guarantee that the learned policy lie in the support region of the source domain dataset and the target domain dataset (such that term (a) and term (b) in Theorem 3.1 can be controlled); (c) No per dataset hyperparameter tuning is required for IQL; (d) IQL consumes less training time than other offline RL methods like CQL.
> > >
> > > Nevertheless, we agree that it would be interesting to investigate how OTDF behaves if we use another base offline RL algorithm. To that end, we utilize TD3BC [11] as the base algorithm first for OTDF and conduct experiments on some selected tasks. We also utilize TD3BC as the base algorithm for DARA and IGDF. We do not use BCQ since it is also an in-sample learning approach akin to IQL, and it runs much slower than TD3BC. We cannot include all possible experiments and can only report part of them here (kinematic shift tasks and morphology shift tasks). Note that we only replace the base algorithm and do not tune any hyperparameters. We run all methods for 5 independent runs and summarize the average normalized score results in the target domain below. The results in Table 1 and Table 2 clearly show that OTDF enjoys strong performance with TD3BC as the base algorithm (competitive to OTDF with IQL as the base algorithm), and surpasses baselines like DARA and IGDF significantly on numerous tasks.

---

> > > > ### Author Response · Authors · 2024-11-20
> > > > **Author Responses to Reviewer i1Jo (part 4)**
> > > >
> > > > | Source | Target | OTDF (IQL) | OTDF (TD3BC) | DARA (TD3BC) | IGDF (TD3BC) |
> > > > | ---- | ---- | :---: | :---: | :---: | :---: |
> > > > | half-m | medium | 40.2$\pm$0.0 | 40.7$\pm$1.3 | 30.7$\pm$9.8 | 29.7$\pm$7.8 |
> > > > | half-m | medium-expert | 10.1$\pm$4.0 | 13.0$\pm$4.8 | 16.2$\pm$4.9 | 5.2$\pm$1.8 |
> > > > | half-m-r | medium | 37.8$\pm$2.1 | 41.7$\pm$0.3 | 20.7$\pm$10.3 | 21.5$\pm$4.6 |
> > > > | half-m-r | medium-expert | 9.7$\pm$2.0 | 21.8$\pm$6.6 | 7.4$\pm$3.0 | 11.5$\pm$5.3 |
> > > > | half-m-e | medium | 30.7$\pm$9.6 | 40.3$\pm$2.0 | 27.8$\pm$3.2 | 28.2$\pm$6.8 |
> > > > | hopp-m | medium | 65.6$\pm$1.9 | 59.2$\pm$8.7 | 33.0$\pm$18.8 | 26.9$\pm$17.9 |
> > > > | hopp-m | medium-expert | 55.4$\pm$25.1 | 47.4$\pm$19.3 | 32.1$\pm$27.1 | 35.4$\pm$22.9 |
> > > > | hopp-m-r | medium | 35.5$\pm$12.2 | 64.5$\pm$2.8 | 44.0$\pm$13.1 | 46.2$\pm$7.7 |
> > > > | hopp-m-r | medium-expert | 47.5$\pm$14.6 | 54.7$\pm$23.4 | 19.1$\pm$7.9 | 22.0$\pm$18.4 |
> > > > | hopp-m-e | medium | 65.3$\pm$2.4 | 64.8$\pm$1.8 | 51.1$\pm$15.0 | 60.1$\pm$5.7 |
> > > > | walk-m | medium | 49.6$\pm$18.0 | 55.1$\pm$10.1 | 25.9$\pm$8.6 | 40.0$\pm$15.1 |
> > > > | walk-m | medium-expert | 43.5$\pm$16.4 | 24.8$\pm$6.1 | 26.4$\pm$4.8 | 14.0$\pm$13.0 |
> > > > | walk-m-r | medium | 49.7$\pm$9.7 | 36.4$\pm$16.3 | 18.0$\pm$4.2 | 21.3$\pm$7.2 |
> > > > | walk-m-r | medium-expert | 55.9$\pm$17.1 | 26.6$\pm$15.2 | 19.1$\pm$11.6 | 24.0$\pm$11.5 |
> > > > | walk-m-e | medium | 44.6$\pm$6.0 | 40.7$\pm$14.7 | 29.2$\pm$14.0 | 36.0$\pm$13.4 |
> > > > | ant-m | medium | 55.4$\pm$0.0 | 51.6$\pm$5.5 | 23.2$\pm$4.0 | 29.1$\pm$4.8 |
> > > > | ant-m | medium-expert | 60.7$\pm$3.6 | 49.8$\pm$7.4 | 30.7$\pm$8.1 | 26.7$\pm$6.9 |
> > > > | ant-m-r | medium | 52.8$\pm$4.4 | 51.4$\pm$3.3 | 24.4$\pm$5.8 | 26.0$\pm$5.0 |
> > > > | ant-m-r | medium-expert | 54.2$\pm$5.2 | 53.2$\pm$6.0 | 30.5$\pm$6.7 | 34.0$\pm$8.0 |
> > > > | ant-m-e | medium | 50.2$\pm$4.3 | 52.4$\pm$4.8 | 23.5$\pm$5.1 | 18.0$\pm$11.7 |
> > > > | Total |  | 913.9 | 890.1 | 533.0 | 555.8 |
> > > >
> > > > Table 1. Results on the kinematic tasks with TD3BC as the backbone for different cross-domain offline RL methods. half=halfcheetah, hopp=hopper, walk=walker2d, m=medium, r=replay, e=expert. We report the average normalized scores in the target domain in conjunction with the standard deviations.
> > > >
> > > > | Source | Target | OTDF (IQL) | OTDF (TD3BC) | DARA (TD3BC) | IGDF (TD3BC) |
> > > > | ---- | ---- | :---: | :---: | :---: | :---: |
> > > > | half-m | medium | 39.1$\pm$2.3 | 40.3$\pm$1.8 | 40.0$\pm$0.9 | 40.1$\pm$0.5 |
> > > > | half-m | medium-expert | 35.6$\pm$0.7 | 35.5$\pm$1.0 | 24.4$\pm$8.2 | 28.5$\pm$1.2 |
> > > > | half-m-r | medium | 40.0$\pm$1.2 | 42.4$\pm$0.5 | 27.7$\pm$6.3 | 29.6$\pm$7.0 |
> > > > | half-m-r | medium-expert | 34.4$\pm$0.7 | 34.4$\pm$4.1 | 16.0$\pm$4.0 | 14.9$\pm$5.2 |
> > > > | half-m-e | medium | 41.4$\pm$0.3 | 39.9$\pm$2.5 | 38.0$\pm$2.4 | 38.4$\pm$1.7 |
> > > > | hopp-m | medium | 11.0$\pm$0.9 | 27.3$\pm$5.4 | 13.2$\pm$0.3 | 13.2$\pm$0.2 |
> > > > | hopp-m | medium-expert | 12.6$\pm$0.8 | 20.6$\pm$14.3 | 12.3$\pm$1.4 | 11.3$\pm$2.6 |
> > > > | hopp-m-r | medium | 8.7$\pm$2.8 | 14.8$\pm$4.8 | 9.3$\pm$1.9 | 9.5$\pm$2.1 |
> > > > | hopp-m-r | medium-expert | 9.7$\pm$2.7 | 18.2$\pm$12.2 | 10.6$\pm$1.0 | 10.3$\pm$0.6 |
> > > > | hopp-m-e | medium | 7.9$\pm$3.2 | 27.5$\pm$12.9 | 15.6$\pm$3.5 | 13.0$\pm$1.7 |
> > > > | walk-m | medium | 50.5$\pm$5.8 | 3.1$\pm$0.1 | 6.9$\pm$2.4 | 16.5$\pm$17.9 |
> > > > | walk-m | medium-expert | 44.3$\pm$23.8 | 52.2$\pm$21.9 | 9.4$\pm$3.3 | 12.5$\pm$1.6 |
> > > > | walk-m-r | medium | 37.4$\pm$5.1 | 24.0$\pm$21.1 | 21.7$\pm$12.0 | 19.6$\pm$10.8 |
> > > > | walk-m-r | medium-expert | 33.8$\pm$6.9 | 33.4$\pm$8.1 | 1.3$\pm$0.7 | 5.8$\pm$2.5 |
> > > > | walk-m-e | medium | 49.9$\pm$4.6 | 9.7$\pm$13.2 | 8.0$\pm$2.0 | 6.9$\pm$1.5 |
> > > > | ant-m | medium | 39.4$\pm$1.7 | 42.8$\pm$0.3 | 32.5$\pm$3.4 | 20.2$\pm$18.3 |
> > > > | ant-m | medium-expert | 58.3$\pm$8.9 | 55.2$\pm$8.9 | 33.6$\pm$4.3 | 27.0$\pm$7.7 |
> > > > | ant-m-r | medium | 41.2$\pm$0.9 | 41.9$\pm$0.9 | 34.4$\pm$2.4 | 34.9$\pm$1.4 |
> > > > | ant-m-r | medium-expert | 50.8$\pm$4.5 | 56.6$\pm$3.8 | 31.3$\pm$2.1 | 33.0$\pm$4.9 |
> > > > | ant-m-e | medium | 39.9$\pm$2.9 | 42.0$\pm$1.1 | 33.1$\pm$6.9 | 29.1$\pm$6.8 |
> > > > | Total | | 685.9 | 661.8 | 419.3 | 414.3 |
> > > >
> > > > Table 2. Results on the morphology tasks with TD3BC as the backbone for different cross-domain offline RL methods. We report the average normalized scores in the target domain in conjunction with the standard deviations.
> > > >
> > > > Furthermore, we adopt CQL [12] as the backbone algorithm for OTDF. Since CQL is very slow, we can only combine CQL with OTDF and summarize the results on some kinematic and morphology tasks in Table 3 and Table 4. We find that the performance of OTDF is comparatively inferior when adopting CQL as the backbone. The reason can be that we adopt a default hyperparameter setup for CQL (e.g., the penalty coefficient $\alpha$ is set to be 10.0 for all tasks, which can be large). It turns out that the base algorithm can have a large impact on the performance of OTDF and we recommend using IQL by default.

---

> > > > > ### Author Response · Authors · 2024-11-20
> > > > > **Author Responses to Reviewer i1Jo (part 5)**
> > > > >
> > > > > | Source | Target | OTDF (IQL) | OTDF (CQL) |
> > > > > | ---- | ---- | :---: | :---: |
> > > > > | half-m | medium | 40.2$\pm$0.0 | 22.1$\pm$12.2 |
> > > > > | half-m | medium-expert | 10.1$\pm$4.0 | 12.1$\pm$12.6 |
> > > > > | half-m-r | medium | 37.8$\pm$2.1 | 26.6$\pm$11.8 |
> > > > > | half-m-r | medium-expert | 9.7$\pm$2.0 | 29.8$\pm$5.0 |
> > > > > | hopp-m | medium | 65.6$\pm$1.9 | 65.4$\pm$0.7 |
> > > > > | hopp-m | medium-expert | 55.4$\pm$25.1 | 25.0$\pm$18.1 |
> > > > > | hopp-m-r | medium | 35.5$\pm$12.2 | 37.5$\pm$13.9 |
> > > > > | hopp-m-r | medium-expert | 47.5$\pm$14.6 | 39.0$\pm$15.5 |
> > > > > | walk-m | medium | 49.6$\pm$18.0 | 32.7$\pm$8.9 |
> > > > > | walk-m | medium-expert | 43.5$\pm$16.4 | 38.5$\pm$17.4 |
> > > > > | walk-m-r | medium | 49.7$\pm$9.7 | 25.5$\pm$9.3 |
> > > > > | walk-m-r | medium-expert | 55.9$\pm$17.1 | 27.9$\pm$10.0 |
> > > > > | ant-m | medium | 55.4$\pm$0.0 | 49.3$\pm$6.0 |
> > > > > | ant-m | medium-expert | 60.7$\pm$3.6 | 56.7$\pm$5.1 |
> > > > > | ant-m-r | medium | 52.8$\pm$4.4 | 53.1$\pm$5.4 |
> > > > > | ant-m-r | medium-expert | 54.2$\pm$5.2 | 50.7$\pm$4.7 |
> > > > > | Total |  | 723.6 | 591.9 |
> > > > >
> > > > > Table 3. Results on the kinematic tasks with CQL as the backbone algorithm. We report the average normalized scores in the target domain and $\pm$ captures the standard deviations.
> > > > >
> > > > > | Source | Target | OTDF (IQL) | OTDF (CQL) |
> > > > > | ---- | ---- | :---: | :---: |
> > > > > | half-m | medium | 39.1$\pm$2.3 | 40.2$\pm$0.9 |
> > > > > | half-m | medium-expert | 35.6$\pm$0.7 | 34.1$\pm$1.8 |
> > > > > | half-m-r | medium | 40.0$\pm$1.2 | 41.5$\pm$0.3 |
> > > > > | half-m-r | medium-expert | 34.4$\pm$0.7 | 29.7$\pm$5.7 |
> > > > > | hopp-m | medium | 11.0$\pm$0.9 | 13.4$\pm$0.5 |
> > > > > | hopp-m | medium-expert | 12.6$\pm$0.8 | 13.7$\pm$0.5 |
> > > > > | hopp-m-r | medium | 8.7$\pm$2.8 | 4.3$\pm$3.0 |
> > > > > | hopp-m-r | medium-expert | 9.7$\pm$2.7 | 12.1$\pm$4.6 |
> > > > > | walk-m | medium | 50.5$\pm$5.8 | 36.6$\pm$10.5 |
> > > > > | walk-m | medium-expert | 44.3$\pm$23.8 | 24.1$\pm$3.7 |
> > > > > | walk-m-r | medium | 37.4$\pm$5.1 | 35.4$\pm$7.9 |
> > > > > | walk-m-r | medium-expert | 33.8$\pm$6.9 | 12.4$\pm$5.5 |
> > > > > | ant-m | medium | 39.4$\pm$1.7 | 39.8$\pm$2.6 |
> > > > > | ant-m | medium-expert | 58.3$\pm$8.9 | 53.0$\pm$3.0 |
> > > > > | ant-m-r | medium | 41.2$\pm$0.9 | 40.7$\pm$1.9 |
> > > > > | ant-m-r | medium-expert | 50.8$\pm$4.5 | 50.7$\pm$1.7 |
> > > > > | Total | | 546.8 | 481.7 |
> > > > >
> > > > > Table 4. Results on the morphology tasks with CQL as the backbone algorithm. We report the average normalized scores in the target domain and $\pm$ captures the standard deviations.
> > > > >
> > > > > Hopefully, these can resolve the concerns. If there is still something unclear, please let us know!
> > > > >
> > > > > [1] Dara: Dynamics-aware reward augmentation in offline reinforcement learning
> > > > >
> > > > > [2] Contrastive representation for data filtering in cross-domain offline reinforcement learning
> > > > >
> > > > > [3] Beyond ood state actions: Supported cross-domain offline reinforcement learning
> > > > >
> > > > > [4] Bridging Offline Reinforcement Learning and Imitation Learning: A Tale of Pessimism
> > > > >
> > > > > [5] Off-policy deep reinforcement learning without exploration
> > > > >
> > > > > [6] Supported policy optimization for offline reinforcement learning
> > > > >
> > > > > [7] Plas: Latent action space for offline reinforcement learning
> > > > >
> > > > > [8] When to trust your simulator: Dynamics-aware hybrid offline-and-online reinforcement learning
> > > > >
> > > > > [9] Off-Dynamics Reinforcement Learning via Domain Adaptation and Reward Augmented Imitation
> > > > >
> > > > > [10] Return Augmented Decision Transformer for Off-Dynamics Reinforcement Learning
> > > > >
> > > > > [11] A minimalist approach to offline reinforcement learning
> > > > >
> > > > > [12] Conservative q-learning for offline reinforcement learning

---

> > > > > > ### Author Response · Authors · 2024-11-24
> > > > > > **Following up with Reviewer i1Jo**
> > > > > >
> > > > > > Dear Reviewer i1Jo, thanks for your thoughtful review. As the author-reviewer discussion period is near its end, we wonder if our rebuttal addresses your concerns. Please let us know if there is anything unclear!

---

> > > > > > > ### Author Response · Authors · 2024-11-28
> > > > > > >
> > > > > > > Dear Reviewer i1Jo, we deeply appreciate your thoughtful review and your time, and hope that our response can address your concerns. We would like to kindly confirm if you still have any concerns or questions. We are more than happy to have further discussions with the reviewer if possible!

---

### Author Response · Authors · 2024-11-20
**Revision Summary**

Dear reviewers,

Thanks for your time in reviewing our paper and valuable advice in making our paper better. We have uploaded a revision of our paper where we:

- cited some related works on off-dynamics RL recommended by Reviewer i1Jo
- fixed all the tables in our paper to ensure that they look pleasing and are not too wide, as recommended by Reviewer 6ito
- clarified the notation of $\mu_{t,t^\prime}$ as concerned by Reviewer vhAr

All modifications are highlighted in **green**. We welcome any suggestions from the reviewer and are pleased to have further discussions with the reviewers.

Best,

Submission 4658 authors

---

### Meta-Review · Area_Chair_hAUm · 2024-12-23

**Metareview:**

This paper develops a procedure to sub-sample source data collected from many different tasks to build a reinforcement-learned (RL) policy on the target task using a small amount of offline data. The sub-sampling is conducted using optimal transport to calculate a propensity score for the behavior policy on the target data. The paper discusses results of this approach on standard offline RL problems using MuJoCo simulation environments.

This paper received two substantial reviews. The authors have done a remarkable job of addressing the concerns of the reviewers in the rebuttal, through new experiments, and detailed explanations. I recommend this paper for an accept. I would suggest the authors to incorporate the discourse from the rebuttal phase into the main paper. I would also suggest them to be more careful in making bold the numbers in Table 1 and 2, it is reasonable to make an entry bold only if it is statistically better than the other columns (e.g., with a p-value of 0.05). Since both the expert and random policies come with very high variance in these environments, the normalized score reported in these tables has a large variance---one could compute it more rigorously.

**Additional Comments On Reviewer Discussion:**

Reviewer i1Jo was concerned about the computational overhead of building a conditional VAE to compute the propensity score of the behavior policy for the target data, ablation studies, and related work. The authors have addressed these comments elaborately in their rebuttal.

Reviewer 6ito was concerned as to why the authors use source data of a substantially different quality than the target data (e.g., source data is from a random behavior policy and target data is from an expert). The authors have addressed this concern satisfactorily. The reviewer also pointed out to an older work that exactly uses optimal transport for cross-domain adaptation in the context of imitation learning---the present paper is (marginally, in my opinion) different because it is working with offline reinforcement learning. Altogether, the review was satisfied with this response and raised their score.

Reviewer vhAr provided a very superficial review with a low confidence and a high score. I suggest ignoring this review.

It was difficult to get the reviewers to engage more with this paper despite some efforts. But the authors have provided a comprehensive rebuttal which helps understand the merits and deficiencies of this work. My recommendation is also based on my own reading of the paper.

---

### Decision · Program_Chairs · 2025-01-22

Accept (Poster)